# FAIR CLUSTERING IN THE SLIDING WINDOW MODEL

**Vincent Cohen-Addad**
Google
cohenaddad@google.com

**Shaofeng H.-C. Jiang**
School of Computer Science, Peking University
shaofeng.jiang@pku.edu.cn

**Qiaoyuan Yang**
School of Computer Science, Peking University
qiaoyuanyang@stu.pku.edu.cn

**Yubo Zhang**
School of Computer Science, Peking University
zhangyubo18@stu.pku.edu.cn

**Samson Zhou**
Texas A&M University
samsonzhou@gmail.com

## ABSTRACT

We study streaming algorithms for proportionally fair clustering, a notion originally suggested by Chierichetti et al. (2017), in the sliding window model. We show that although there exist efficient streaming algorithms in the insertion-only model, surprisingly no algorithm can achieve finite ratio without violating the fairness constraint in sliding window. Hence, the problem of fair clustering is a rare separation between the insertion-only streaming model and the sliding window model. On the other hand, we show that if the fairness constraint is relaxed by a multiplicative $(1 + \varepsilon)$ factor, there exists a $(1 + \varepsilon)$-approximate sliding window algorithm that uses $\text{poly}(k\varepsilon^{-1} \log n)$ space. This achieves essentially the best parameters (up to degree in the polynomial) provided the aforementioned lower bound. We also implement a number of empirical evaluations on real datasets to complement our theoretical results.

## 1 INTRODUCTION

Clustering is a fundamental technique used to identify meaningful patterns and structures within data. Typically, the objective of clustering involves grouping similar data points together into distinct clusters. Due to extensive research conducted over the years, clustering has become a well-established and deeply understood field. In the standard notion of clustering, the input is a set $P$ of $n$ points in a space equipped with metric $\text{dist}$, a cluster parameter $k > 0$, and an exponent $z > 0$ that is a positive integer and the objective is to minimize the quantity $\min_{C,\Gamma:|C|=k} \sum_{p \in P} \min_{c \in C} \text{dist}(p, c)^z$. The problem is called $(k, z)$-clustering when the metric space is $\mathbb{R}^d$ and $\text{dist}$ is the Euclidean distance and in particular, the problem is called $k$-median for $z = 1$ and $k$-means for $z = 2$. Note that the center set $C$ implicitly partitions $P$ by grouping together the points $p \in P$ closest to each center $c \in C$.

**Fair clustering.** When applied to user data, traditional clustering methods may produce biased outputs, leading to discriminatory outcomes that can perpetuate social inequalities in the downstream applications (Dwork et al., 2012; Chhabra et al., 2021). For example, clustering algorithms used in hiring or lending tasks may inadvertently discriminate against certain groups based on factors such as race, gender, or socioeconomic status. This is particularly problematic in domains where clustering is used to make decisions with significant consequences, such as healthcare, education, and finance. Hence, there has been a large line of active work focusing on various notions of fairness for clustering; we defer an overview of this work to Section 1.2.

In $(\alpha, \beta)$-fair clustering, each item $p \in P$ is associated with a subpopulation (often also called a group) $j \in [\ell]$ across $\ell$ possible subpopulations and additional constraints $0 \le \alpha_j \le \beta_j \le 1$ for each $i \in [\ell]$. Then the goal is to minimize the clustering objective over all possible clusters where each cluster has at least $\alpha_j$ fraction and no more than $\beta_j$ fraction of items in group $L$. Thus,

these assignment constraints are designed to capture fairness by ensuring that no subpopulation is under-represented or over-represented in each cluster.

**The sliding window model.** The sliding window model (Datar et al., 2002), which focuses on analyzing recent data points rather than the entire dataset, offers a unique perspective on this challenge. In many real-world applications, recent data is more relevant and informative than historical data (Babcock et al., 2002; Manku & Motwani, 2012; Papapetrou et al., 2015; Wei et al., 2016). For example, in social media analysis, understanding current trends and discussions requires focusing on the most recent posts. In these scenarios, the sliding window model can provide a more dynamic and responsive approach to clustering.

The sliding window model is particularly relevant for applications in which computation must be restricted to data received after a certain time. Laws regarding data privacy, including the sweeping General Data Protection Regulation (GDPR), explicitly require that companies must not retain certain user information beyond a specified time. Consequently, Facebook retains user search data for 6 months (Facebook), the Apple retains user information for 3 months (Apple), and Google stores browser information for up to 9 months (Google). The sliding window model captures these retention policies with the appropriate setting of the window parameter $W$ and thus has been considered across a wide range of applications (Lee & Ting, 2006a;b; Braverman & Ostrovsky, 2007; Chen et al., 2016; Datar & Motwani, 2016; Epasto et al., 2017; Braverman et al., 2018; 2021; Woodruff & Zhou, 2021; Jayaram et al., 2022; Blocki et al., 2023).

However, the sliding window model also introduces new challenges for fairness. As data streams in continuously, the composition of the window can change rapidly, potentially leading to biased or discriminatory outcomes. Therefore, it is essential to develop clustering algorithms that are not only accurate but also fair in the context of the sliding window model. Formally, points arrive sequentially in the sliding window model and the dataset $P$ is implicitly defined as the last $n$ points of the stream. The goal is to perform fair clustering using space sublinear in the size $n$ of the dataset.

## 1.1 OUR RESULTS

In this work, we study fair clustering in the sliding window model. In particular, our goal is to achieve $(1 + \varepsilon)$-multiplicative approximation to the optimal fair clustering for the dataset implicitly defined by the data stream in the sliding window model, while using the minimal amount of space (and then minimizing the update time as a secondary priority). Surprisingly, there is no existing work on fair clustering in the sliding window model.

On the other hand, recent work has achieved $(1 + \varepsilon)$-approximation for the standard formulation of $(k, z)$-clustering in the sliding window model using $\frac{k}{\min(\varepsilon^4, \varepsilon^{2+z})}$ polylog $\frac{n\Delta}{\varepsilon}$ words of space (Woodruff et al., 2023), which was shown to be space-optimal up to factors polylogarithmic in the input size $n$ and the aspect ratio $\Delta$ (Cohen-Addad et al., 2022; 2023; Huang et al., 2024). In particular, we can assume without loss of generality that each coordinate of each point can be represented in $\log \Delta$ bits of space, so that after normalization, all points lie within the grid $[\Delta]^d = \{1, 2, \ldots, \Delta\}^d$. Similarly, results by Braverman et al. (2022) can be adapted using the well-known merge-and-reduce framework to achieve a $(1 + \varepsilon)$-approximation streaming algorithm for fair clustering using $\frac{k}{\min(\varepsilon^4, \varepsilon^{2+z})}$ polylog $\frac{n\Delta}{\varepsilon}$ words of space, which is also space-optimal up to polylogarithmic factors (Cohen-Addad et al., 2022; Huang et al., 2024). Therefore, we ask:

> Can we achieve similar space-optimal results for fair clustering in the sliding window model?

Surprisingly, our first result is a strong impossibility result:

**Theorem 1.1.** *Any algorithm for fair $(k, z)$-clustering in the sliding window model that either achieves any multiplicative approximation or additive $\frac{\Delta}{2} - 1$ error, with probability at least $\frac{2}{3}$, must use $\Omega(n)$ space.*

The proof of Theorem 1.1 utilizes tools from communication complexity and is fairly standard. Specifically, we show that fair clustering in the sliding window model can solve a version of Augmented Index, where one player has a binary string $x \in \{0, 1\}^{2n}$ of length $2n$ and weight $n$, while the second player has an index $i$ and must guess the value of $x_i$, given the values of $x_1, \ldots, x_{i-1}$,

as well as a message passed from the first player. The main intuition is that the two players can copy $x_1, \ldots, x_{i-1}$ to the end of the stream, as well as a guess $x_i = 0$, so that the sliding window contains precisely the elements $x_{i+1}, \ldots, x_{2n}, x_1, x_2, \ldots, x_{i-1}, 0$, as well as $n$ additional points at the origin and $n$ additional points at the point $1$ on the real line, which the players subsequently insert. Now if the guess $x_i = 0$ is correct, this window will contain precisely $2n$ points that zero and $2n$ points that are one. Otherwise, it will contain $2n - 1$ points that are one and $2n + 1$ points that are zero. Hence by setting the fairness constraint to require half of the points at each center to be from the last $2n$ points, and half of the points from the first $2n$ points, the fair clustering objective is zero if and only if the guess $x_i = 0$ is correct. Otherwise, the objective is nonzero, which allows any finite multiplicative approximation to fair clustering to distinguish between these two cases. The lower bound then follows from known communication lower bounds of the Augmented Index problem. We defer the formal proof to Appendix A. We now discuss the main implications of Theorem 1.1.

Whereas our goal was to achieve a $(1 + \varepsilon)$-approximation for fair clustering in $\mathrm{poly}\left(k, \frac{1}{\varepsilon}, \log(n\Delta)\right)$ space, Theorem 1.1 states that *any* multiplicative approximation requires linear space, even an arbitrarily large multiplicative approximation, e.g., polynomial or exponential in $n$.

Due to the aforementioned results, Theorem 1.1 shows several separations. Firstly, it shows a separation between fair clustering and the standard clustering formulation in the sliding window model, due to the sliding window algorithm for $(k, z)$-clustering that achieves $(1 + \varepsilon)$-approximation using $\frac{k}{\min(\varepsilon^4, \varepsilon^{2+z})}$ polylog $\frac{n}{\varepsilon}$ words of space (Woodruff et al., 2023). Secondly, it shows a separation between the insertion-only streaming model and the sliding window model for fair clustering, due to the streaming algorithm for $(k, z)$-clustering that achieves $(1 + \varepsilon)$-approximation using $\frac{k}{\min(\varepsilon^4, \varepsilon^{2+z})}$ polylog $\frac{n}{\varepsilon}$ words of space (Braverman et al., 2022). Theorem 1.1 therefore has an important informal message:

> Fairness and recency are not compatible in sublinear space!

The natural follow-up question is, what actually can be achieved toward some notion of fair clustering in the sliding window model? To that end, we show that there exist sublinear-space algorithms for fair clustering in the sliding window model if we permit a small slackness on the fairness constraints:

**Theorem 1.2.** *There is a sliding-window streaming algorithm that returns a center set $C$ after every insertion operation, such that with high probability, there exists a $((1 - \varepsilon)\alpha, (1 + \varepsilon)\beta)$-fair $k$-clustering whose cost with center set $C$ is at most $1 + \varepsilon$ times the optimal $(\alpha, \beta)$-fair clustering on the sliding window. This algorithm uses space $L \cdot \mathrm{poly}(\varepsilon^{-1} kd \log \Delta)$ where $L \leq 2^\ell$ is the number of distinct combinations of groups that any point may belong to.*

We remark that the guarantees of our algorithm are slightly stronger than those stated in Theorem 1.2. In particular, not only does our algorithm produce a $(1 + \varepsilon)$-approximation to the optimal fair clustering, but it also produces a $(1 + \varepsilon)$-coreset, i.e., a dataset that can be used to obtain a $(1 + \varepsilon)$-approximation to the cost of any query consisting of $k$ centers under the fairness constraints. To that end, we further remark that the space used by Theorem 1.2 matches the best known bounds for offline coreset constructions up to polylogarithmic factors (Braverman et al., 2022).

**Experimental evaluations.** Although the $(1 + \varepsilon)$ ratio which our Theorem 1.2 achieves is powerful, it is mostly theoretical since the running time must be exponential because of APX-hardness. However, our algorithm actually maintains a *coreset* of the sliding window whose construction is very efficient. Hence, we plug in a practically efficient downstream approximation algorithm by Backurs et al. (2019) to achieve better empirical performance. We validate the performance of our algorithm, compared with three baselines, on five real datasets. Our algorithm offers overall best tradeoff regarding , time/space and accuracy compared with all baselines.

## 1.2 RELATED WORK

In this section, we describe a number of related works, both in the context of fair clustering and sliding window algorithms for clustering.

**Fairness.** Fairness in algorithmic design has recently received significant attention (Kleinberg et al., 2017; Huang et al., 2019; Chen et al., 2024; Song et al., 2024). Chierichetti et al. (2017) initiated the study of fairness in clustering, specifically in the context of *disparate impact*, which demands that

any "protected" subpopulation must receive similar representation in the decision-making process, in particular by the output of an algorithm. Although Chierichetti et al. (2017) initially considered two classes of populations, subsequent works generalized to multiple subpopulations by Rösner & Schmidt (2018), as well as different clustering objectives (Ahmadian et al., 2019; Bandyapadhyay et al., 2019; Bercea et al., 2019; Bera et al., 2019b; Kleindessner et al., 2019b; Ahmadian et al., 2020b;a; Esmaeili et al., 2020; 2021; Böhm et al., 2021; Schwartz & Zats, 2022; Ahmadian & Negahbani, 2023). A similar notion studied clustering where each cluster required a certain number of representatives from each subpopulation, rather than a certain fraction (Anegg et al., 2022; Jia et al., 2022).

Other proposed notions of fairness include (1) social fairness, where the clustering cost is considered across subpopulations and then minimized (Jones et al., 2020; Ghadiri et al., 2021; Makarychev & Vakilian, 2021), (2) individual fairness, where each point should have a center within a reasonably close distance that is a function of the overall dataset (Jung et al., 2020; Mahabadi & Vakilian, 2020; Negahbani & Chakrabarty, 2021; Vakilian & Yalçiner, 2022), (3) representative fairness, where the number of centers of each class is constrained (Kleindessner et al., 2019a; Angelidakis et al., 2022), and (4) a number of other notions (Micha & Shah, 2020; Abbasi et al., 2021; Hotegni et al., 2023; Gupta et al., 2023). However, as we focus on disparate impact in this work, these notions of fairness are somewhat orthogonal to the main subject of our study.

**Clustering in data streams and the sliding window model.** While a general framework for sliding window algorithms are histogram-based approaches (Datar et al., 2002; Braverman & Ostrovsky, 2007), the clustering objective is not smooth and thus not suitable for these frameworks. Hence, there has been an active line of work studying $(k, z)$-clustering in the sliding window model. Babcock et al. (2003) first introduced an algorithm for $k$-median clustering in the sliding window model that used $\mathcal{O}\left(\frac{k}{\varepsilon^4} m^{2\varepsilon} \log^2 m\right)$ words of space and gave a $2^{\mathcal{O}(1/\varepsilon)}$-multiplicative approximation algorithm, where $m$ is the size of the window and $\varepsilon \in \left(0, \frac{1}{2}\right)$ is an input parameter. Braverman et al. (2015) subsequently gave a bicriteria algorithm for the $k$-median problem in the sliding window model that achieved an $\mathcal{O}(1)$-multiplicative approximation using $k^2 \operatorname{polylog}(m)$ space, but at the cost of $2k$ centers. Braverman et al. (2016) achieved the first $\operatorname{poly}(k \log m)$ space algorithm for $k$-median and $k$-means on sliding windows, achieving an $\mathcal{O}(1)$-approximation using $\mathcal{O}\left(k^3 \log^6 m\right)$ space. Afterwards, Borassi et al. (2020) achieved a linear dependency in $k$, giving an $\mathcal{O}(1)$-approximation for $k$-clustering using $k \operatorname{polylog}(m, \Delta)$ space. Epasto et al. (2022) introduced the first $(1 + \varepsilon)$-approximation algorithm for $(k, z)$-clustering using $\frac{(kd + d^C)}{\varepsilon^3} \operatorname{polylog}\left(m, \Delta, \frac{1}{\varepsilon}\right)$ words of space, for some constant $C \geq 7$, which was subsequently optimized by Woodruff et al. (2023) to $\frac{k}{\min(\varepsilon^4, \varepsilon^{2+z})} \operatorname{polylog} \frac{m\Delta}{\varepsilon}$ words of space.

It should noted that all of these results consider the standard formulation of $(k, z)$-clustering in the sliding window model. That is, no previous works considered fair clustering in the sliding window model. The most relevant works are offline $(1 + \varepsilon)$-coreset constructions by Braverman et al. (2022) for fair clustering that sample $\frac{k}{\min(\varepsilon^4, \varepsilon^{2+z})} \operatorname{polylog} \frac{n\Delta}{\varepsilon}$ points, and can be adapted to the insertion-only streaming model at the cost of multiplicative factors polylogarithmic in $n$, using a standard merge-and-reduce technique. As this coreset size matches the bound for $(k, z)$-clustering in the sliding window model, one might hope to achieve similar bounds for fair clustering in the sliding window model. Unfortunately, our results show that is not the case.

## 2 PRELIMINARIES

**Notations.** We assume the input comes from $[\Delta]^d$ for some integer $\Delta \geq 1$, and let $\mathcal{C} \subseteq \mathbb{R}^d$ be the candidate center set. In our analysis, many statements also hold for general $\mathbb{R}^d$, but we may still state them for $[\Delta]^d$ for consistency of presentation. Throughout, when we talk about a point set in $\mathbb{R}^d$, we assume each point is associated with a unique time step. Importantly, we use this time step to identify a point, which means two identical points in $\mathbb{R}^d$ with different time steps are considered different points (and set operations, such as intersection, are performed with respect to the time step). For a general function $f : U \to V$, define $f(Y) := \sum_{y \in Y} f(y)$ (for $Y \subseteq U$ in the domain $U$).

**Definition 2.1** ($(\alpha, \beta)$-fair clustering)**.** *Given $P \subset \mathbb{R}^d$, $\ell$ groups $P_1, \ldots, P_\ell \subseteq P$ and $\alpha, \beta \in [0, 1]^\ell$, a $k$-clustering $(C_1, \ldots, C_k)$, i.e., a $k$-partition of $P$, is called $(\alpha, \beta)$-fair if $\frac{|P_j \cap C_i|}{|C_i|} \in [\alpha_j, \beta_j]$ for every*

$i \in [k], j \in [\ell]$. $(\alpha, \beta)$-*fair* $(k, z)$-CLUSTERING *asks to find an* $(\alpha, \beta)$-*fair clustering* $(C_1, \ldots, C_k)$ *and center set* $C \subset \mathbb{R}^d$ *of* $k$ *points, such that* $\sum_{i \in [k]} \sum_{x \in C_i} (\mathrm{dist}(x, c_i))^z$ *is minimized.*

Notice that there are at most $2^\ell$ possible combinations of groups that a point $p \in P$ may belong to. In practice, this $2^\ell$ upper bound is rarely achieved. Hence, it is often useful to consider the actual number of distinct combinations of groups that a point $p \in P$ may belong to, denoted as $L$ throughout. Namely, $L := |\{W \subseteq [\ell] : \exists p \in P, \text{ such that } \forall j \in W, p \in P_j\}|$.

**Streaming model.** The window size $m$ and the fairness constraints $\alpha, \beta \in \mathbb{R}^\ell$ are given in advance. The input point set $P \subseteq [\Delta]^d$ and the groups $P_1, \ldots, P_\ell \subseteq P$ of $(\alpha, \beta)$-fair clustering is presented as a stream of insertion operations, each consists of a point $p \in P$ and the group IDs that $p$ belongs to. The algorithm must return a center set $C \subset \mathbb{R}^d$ after each insertion, which is supposed to be an approximate solution to the dataset defined by the last $m$ operations.

A weighted point set $S \subseteq [\Delta]^d$ is a point set $S$ equipped with a weight function $w_S : [\Delta]^d \to \mathbb{R}_+$ such that $\mathrm{supp}(w_S) = S$. Notice that for a weighted point set $S$ and a (weighted or unweighted) point set $T$, set operation $S \cap T$ is performed only on the point set, and the weight should be defined separately. When the context is clear, we drop the subscript $S$ in the weight function $w_S$.

We use a generic notion of assignment-preserving coresets as introduced in Braverman et al. (2022), which we rephrase as follows. We notice that an assignment-preserving coreset is not immediately a coreset that preserves the fair clustering objective, but it is possible to obtain one by taking a union of several assignment-preserving coresets on a partition of the input dataset (and the detailed argument can be found in Section 4, proof of Theorem 1.2).

**Definition 2.2** (Assignment constraints). *Consider some weighted point set* $P \subseteq [\Delta]^d$ *and a center set* $C \subseteq \mathcal{C}$. *An assignment constraint with respect to* $P$ *and* $C$ *is a function* $\Gamma : C \to \mathbb{R}_+$ *such that* $\sum_{p \in P} w_P(p) = \sum_{c \in C} \Gamma(c)$. *An assignment function with respect to* $P$ *and* $C$ *is a function* $\sigma : P \times C \to \mathbb{R}_+$ *such that* $\forall p \in P, \sum_{c \in C} \sigma(p, c) = w_P(p)$. *We call* $\sigma$ *is consistent with* $\Gamma$, *i.e.* $\sigma \sim \Gamma$, *if* $\forall c \in C, \sum_{p \in P} \sigma(p, c) = \Gamma(c)$.

**Definition 2.3** ($(k, z)$-clustering with assignment constraints). *Given a weighted point set* $P \subseteq [\Delta]^d$, *a center set* $C \subseteq \mathcal{C}$ *and an assignment constraint* $\Gamma$, *for every assignment function* $\sigma$ *with respect to* $P, C$ *and consistent with* $\Gamma$, *the cost of* $\sigma$ *is defined as* $\mathrm{cost}^\sigma(P, C, \Gamma) = \sum_{p \in P} \sum_{c \in C} \sigma(p, c) d(p, c)^z$. *The cost function of* $(k, z)$-CLUSTERING *with assignment constraint is*

$$\mathrm{cost}(P, C, \Gamma) = \min_{\sigma : P \times C \to \mathbb{R}_+, \sigma \sim \Gamma} \mathrm{cost}^\sigma(P, C, \Gamma).$$

As in Braverman et al. (2022); Huang et al. (2023), our coreset is defined with respect to assignment constraints, and we aim for coresets that can preserve all assignment constraints simultaneously.

**Definition 2.4** (Assignment-preserving coreset). *Given a weighted point set* $P \subseteq [\Delta]^d$ *and* $0 < \varepsilon < 1$, *an assignment-preserving* $\varepsilon$-*coreset for* $(k, z)$-CLUSTERING *of* $P$ *is a weighted point set* $S \subseteq P$ *such that* $w_S(S) = w_P(P)$, *and that for every center set* $C \subseteq \mathcal{C}$ *and assignment constraint* $\Gamma$, $\mathrm{cost}(S, C, \Gamma) \in (1 \pm \varepsilon) \mathrm{cost}(P, C, \Gamma)$.

## 3 ONLINE ASSIGNMENT-PRESERVING CORESETS

As we mention, our framework is based on that of Woodruff et al. (2023), which reduces the sliding window algorithm to constructing an *online coreset*. In their definition, given a point set $P = \{p_1, p_2, \ldots, p_n\}$, listed in the increasing order of time steps, an online $\varepsilon$-coreset $S$ for $(k, z)$-CLUSTERING is a weighted subset such that every prefix $S \cup \{p_1, p_2, \ldots, p_i\}$ is an $\varepsilon$-coreset for $(k, z)$-CLUSTERING of prefix $\{p_1, p_2, \ldots, p_i\}$.

However, this definition of online coreset does not easily generalize to our setting, and it is even trickier due to our lower bound in Theorem 1.1. Hence, we need to use a slightly relaxed notion, defined in Definition 3.1. Roughly, the prefix property of our online assignment-preserving coresets can tolerate a $1 \pm \varepsilon$ relative error in the weights. This relaxed guarantee leads to an efficient coreset bound, stated in Lemma 3.2, which is the main claim of this section.

**Definition 3.1** (Online assignment-preserving coreset). *Given a weighted point set* $P = \{p_1, p_2, \ldots, p_n\}$ *listed in the increasing order of time steps and* $0 < \varepsilon < 1$, *a weighted point*

---

**Algorithm 1** Online assignment-preserving coreset construction

---

**procedure** ONLINECORESET($P, \varepsilon, \delta$)
    Compute an $(\alpha, \beta)$-approximate solution $Q = \{q_1, q_2, \ldots, q_\ell\} \subseteq P$ for $(k, z)$-clustering of $P$
  ▷using e.g., Meyerson sketch, where $\alpha = O(2^z)$, $\beta = O(2^z \operatorname{poly}(\log \Delta))$
    $S \leftarrow \varnothing, \varepsilon' \leftarrow \frac{\varepsilon}{2^z \alpha}, \delta' \leftarrow \frac{\delta}{\ell(\lceil \lg(\Delta\sqrt{d}) + 1\rceil)}$
    **for** $i = 1, 2, \ldots, \ell$ **do**
        $Q_i \leftarrow \{p \in P : \operatorname{NN}_Q(p) = q_i\}$  ▷$\operatorname{NN}_Q(p)$ is defined as the nearest neighbor of $p$ in set $Q$
        **for** $j = 0, 1, 2 \ldots, \lceil \lg(\Delta\sqrt{d})\rceil$ **do**
            $P_{i,j} \leftarrow P_i \cap \operatorname{ring}(q_i, 2^{j-1}, 2^j)$
            $S_{i,j} \leftarrow \operatorname{RINGCORESET}(P_{i,j}, \varepsilon', \delta')$
    **return** $\bigcup_{1 \le i \le \ell, 0 \le j \le \lceil \lg(\Delta\sqrt{d})\rceil} S_{i,j}$

---

**Algorithm 2** Coreset for a single ring

---

**procedure** RINGCORESET($P, \varepsilon, \delta$)
    List $P$ in the increasing order of time steps as $p_1, p_2, \ldots, p_n$.
    $\delta' \leftarrow \frac{\delta}{(k/\varepsilon)^{O(kd)}}, T \leftarrow \operatorname{poly}(2^z dk\varepsilon^{-1} \log(w(P)\Delta\varepsilon^{-1}\delta^{-1})), S \leftarrow \varnothing, \operatorname{sum}_0 \leftarrow 0$
    **for** $i = 1, 2, \ldots, n$ **do**
        $\operatorname{sum}_i \leftarrow \operatorname{sum}_{i-1} + w_P(p_i)$
        $\operatorname{prob}_i \leftarrow \min(T \cdot w_P(p_i)/\operatorname{sum}_i, 1)$
        add $p_i$ to $S$ with probability $\operatorname{prob}_i$ and weight $w_S(p_i) \leftarrow \operatorname{prob}_i^{-1} w_P(p_i)$
    **return** $S$

---

*set $S$ is an online assignment-preserving $\varepsilon$-coreset of $P$ for $(k, z)$-CLUSTERING, if $S \subseteq P$ and for every $t \in [n]$, there exists some weighted point set $S_t'$ such that the following holds:*

- *Let $P_t := \{p_1, \ldots, p_t\}$ and $w_{P_t} : p \mapsto w_P(p)$. Then $S_t'$ is an assignment-preserving $\varepsilon$-coreset for the weighted set $P_t$.*
- *Let $S_t := S \cap P_t$ and $w_{S_t} : p \mapsto w_S(p)$. Then $S_t = S_t'$ and for every $p \in S_t$, $w_{S_t}(p) \in (1 \pm \epsilon) w_{S_t'}(p)$.*

**Lemma 3.2.** *There exists an algorithm that takes as input weighted set $P \subseteq [\Delta]^d$ (with unique time steps) of $n$ points, $0 < \varepsilon < 1, z \ge 1$ and integer $k \ge 1$, computes weighted set $S$ with $|S| = \operatorname{poly}(2^z \varepsilon^{-1} kd \log(w(P)\Delta\varepsilon^{-1}))$, such that $S$ is online assignment-preserving $\varepsilon$-coreset for $(k, z)$-CLUSTERING with probability $0.9$. The algorithm uses $\operatorname{poly}(2^z \varepsilon^{-1} kd \log(w(P)\Delta\varepsilon^{-1}))$ space.*

**Algorithm for Lemma 3.2.** The algorithm for our online assignment-preserving coreset is listed in Algorithm 1. This construction is based on a similar sampling-based framework as in Woodruff et al. (2023), which is also widely used in the coreset literature in general (Chen, 2009; Cohen-Addad et al., 2021; Cohen-Addad & Li, 2019; Braverman et al., 2022). In this framework, one starts with computing a bi-criteria approximation $\widehat{C}$ for (unconstrained) $(k, z)$-CLUSTERING, denoted as $\widehat{C} := \{\widehat{c}_1, \ldots, \widehat{c}_t\}$. We say $\widehat{C}$ is an $(\alpha, \beta)$ bi-criteria approximate solution for (unconstrained) $(k, z)$-CLUSTERING if $\operatorname{cost}(P, \widehat{C}) = \sum_{p \in P} \operatorname{dist}(x, \widehat{C})^z \le \alpha \cdot \operatorname{OPT}$ and $|\widehat{C}| \le \beta k$. For our purpose, we use the Meyerson sketch proposed by Borassi et al. (2020) that outputs an $(\mathcal{O}(2^z), \mathcal{O}(2^z \log \eta^{-1} \log \Delta))$ approximation center set $\widehat{C} \subseteq P$ with probability at least $1 - \eta$. Next, take the natural clustering defined by $\widehat{C}$ (according to nearest-neighbor rule), and decompose each cluster into rings centered at $\widehat{c}_i$. The coreset is constructed by simply drawing $\operatorname{poly}(\epsilon^{-1} kd)$ uniform samples from each ring, re-weight, and take the union.

However, it is nontrivial to draw exactly uniform samples while still maintaining the online property, especially in the streaming setting. Hence, in Woodruff et al. (2023) they turn to sample the $i$-th element in a ring with probability $\propto 1/i$ (and it gets more complicated when running on a weighted dataset, which we need). Our algorithm also has this part as a key subroutine, which we call RINGCORESET. The RINGCORESET algorithm is presented in Algorithm 2, and its guarantee is summarized in the following Lemma 3.3.

**Lemma 3.3.** *Consider* $q \in [\Delta]^d$, $r > 0$, $\varepsilon, \delta \in (0,1), k, z \geq 1$, *and weighted set* $P = (p_1, p_2, \ldots, p_n) \subseteq [\Delta]^d \cap \mathrm{ring}(q, r/2, r)$ *listed in the increasing order of time steps. Let* $N := w(P)$ *and* $S$ *be the output of* RINGCORESET$(P, \varepsilon, \delta)$ *(in Algorithm 2). We have* $E[|S|] = \mathrm{poly}(2^z k \varepsilon^{-1} \log(N \Delta \varepsilon^{-1} \delta^{-1}))$, *and with probability at least* $1 - \delta$, *there exists a weighted set* $S'$ *such that*

- $S' = S$,
- *For every* $p \in S$, $w_{S'}(p) \in (1 \pm \varepsilon) w_S(p)$,
- *For every center set* $C \subseteq \mathbb{R}^d, |C| \leq k$ *and every assignment constraint* $\Gamma$ *with respect to* $P, C$, *we have* $|\mathrm{cost}(S', C, \Gamma) - \mathrm{cost}(P, C, \Gamma)| \leq \frac{\varepsilon}{4}(\mathrm{cost}(P, C, \Gamma) + \mathrm{cost}(S', C, \Gamma) + Nr^z)$.

The proof of Lemma 3.3 is postponed to Appendix C, and we first finish the proof of Lemma 3.2 assuming Lemma 3.3 holds. In the following discussion, for every $P \subseteq [\Delta]^d, C \subseteq \mathbb{R}^d, |C| = k$ and assignment constraint $\Gamma$ with respect to $P, C$, we say $\sigma^*$ is the the optimal assignment function of $\mathrm{cost}(P, C, \Gamma)$ if $\sigma^* = \arg\min_{\sigma:P \times C \to \mathbb{R}_{\geq 0}, \sigma \sim \Gamma} \sum_{p \in P} \sum_{c \in C} \sigma(p, c) \mathrm{dist}(p, c)^z$.

*Proof of Lemma 3.2.* By the definition of Algorithms 1 and 2, we have ONLINECORESET$(P) \cap \{p_1, p_2, \ldots, p_i\} = $ ONLINECORESET$(\{p_1, p_2, \ldots, p_i\})$ when the randomness is fixed. Hence it is sufficient to show that for every $P$, there exists some assignment-preserving coreset $S'$ of $P$ such that $S' = S$ and for every $p \in S'$, $w_{S'}(p) \in (1 \pm \varepsilon) w_S(p)$.

Let $\alpha, \ell, \varepsilon', S_{i,j}, P_{i,j}$ follows their definition in Algorithm 1. By applying the union bound on the result of Lemma 3.3, with probability at least $1 - \delta$, for every $1 \leq i \leq \ell, 0 \leq j \leq \lceil \log(\Delta\sqrt{d}) \rceil$, there exists a weighted point set $S'_{i,j}$ satisfying $S'_{i,j} = S_{i,j}, \forall p \in S'_{i,j}, w_{S'_{i,j}}(p) \in (1 \pm \varepsilon) w_{S_{i,j}}(p)$, such that for every center set $C$ and assignment constraint $\Gamma$ with respect to $P_{i,j}, C$, we have

$$\left| \mathrm{cost}(S'_{i,j}, C, \Gamma) - \mathrm{cost}(P_{i,j}, C, \Gamma) \right|$$

$$\leq \frac{\varepsilon'}{4} w(P_{i,j}) r_j^z + \frac{\varepsilon'}{4}(\mathrm{cost}(S'_{i,j}, C, \Gamma) + \mathrm{cost}(P_{i,j}, C, \Gamma))$$

$$\leq \frac{\varepsilon}{4\alpha} \sum_{p \in P_{i,j}} w_P(p) \mathrm{dist}(p, q_i)^z + \frac{\varepsilon}{4}(\mathrm{cost}(S'_{i,j}, C, \Gamma) + \mathrm{cost}(P_{i,j}, C, \Gamma))$$

Let weighted set $S'$ be the union of all $S'_{i,j}$. Let $\sigma^*$ be the optimal assignment for $\mathrm{cost}(P, C, \Gamma)$. We fix the assignment constraint for every $P_{i,j}$ with $\Gamma_{i,j}(c) := \sum_{p \in P_{i,j}} \sigma^*(p, c)$. Define $N_R := \lceil \lg(\Delta\sqrt{d}) \rceil$ as the number of rings. We have $\mathrm{cost}(P, C, \Gamma) = \mathrm{cost}^{\sigma^*}(P, C, \Gamma) = \sum_{i=1}^{\ell} \sum_{j=1}^{N_R} \mathrm{cost}(P_{i,j}, C, \Gamma_{i,j})$ and $\mathrm{cost}(S', C, \Gamma) \leq \sum_{i=1}^{\ell} \sum_{j=1}^{N_R} \mathrm{cost}(S'_{i,j}, C, \Gamma_{i,j})$. Summing up together, we have

$$\mathrm{cost}(S', C, \Gamma) - \mathrm{cost}(P, C, \Gamma)$$

$$\leq \sum_{i=1}^{\ell} \sum_{j=1}^{N_R} \mathrm{cost}(S'_{i,j}, C, \Gamma_{i,j}) - \mathrm{cost}(P_{i,j}, C, \Gamma_{i,j})$$

$$\leq \frac{\varepsilon}{4\alpha} \sum_{i=1}^{\ell} \sum_{p \in P_i} w_P(p) \mathrm{dist}(p, q_i)^z + \frac{\varepsilon}{4}(\mathrm{cost}(S', C, \Gamma) + \mathrm{cost}(P, C, \Gamma))$$

$$\leq \frac{\varepsilon}{4\alpha} \mathrm{cost}_Q + \frac{\varepsilon}{4} \mathrm{cost}(P, C, \Gamma) + \frac{\varepsilon}{4} \mathrm{cost}(S', C, \Gamma)$$

$$\leq \varepsilon \cdot \mathrm{cost}(P, C, \Gamma),$$

where $\mathrm{cost}_Q$ denotes the cost of solution $Q$. By symmetric argument we have $\mathrm{cost}(P, C, \Gamma) - \mathrm{cost}(S', C, \Gamma) \leq \varepsilon \cdot \mathrm{cost}(P, C, \Gamma)$. The expectation size of the coreset can be bounded by

$$E[|S|] = \sum_{i=1}^{\ell} \sum_{j=0}^{N_R} |S_{i,j}| \leq \beta k \cdot \lg(\Delta\sqrt{d}) \cdot \mathrm{poly}(2^z dk\varepsilon^{-1} \log(w(P)\Delta\varepsilon^{-1}\delta^{-1}))$$

$$= \mathrm{poly}(2^z dk\varepsilon^{-1} \log(w(P)\Delta\varepsilon^{-1}\delta^{-1})).$$

In summary, when we run ONLINECORESET$(P, \varepsilon, 0.01)$, it returns an online assignment-preserving $\varepsilon$-coreset $S$ such that $|S| = \mathrm{poly}(2^z dk\varepsilon^{-1} \log(w(P)\Delta\varepsilon^{-1}\delta^{-1}))$ with probability 0.9. □

Table 1: Specifications of datasets

| dataset | size | $d$ | attribute | window size | coreset size (benchmark) | coreset size (ours) |
|---------|------|-----|-----------|-------------|--------------------------|---------------------|
| Adult | 50k | 6 | gender | 500 | 231 | 150 |
| Bank | 45k | 10 | marital | 500 | 234 | 150 |
| Diabetes | 100k | 8 | gender | 1000 | 485 | 300 |
| Athlete | 200k | 3 | gender | 2000 | 1058 | 750 |
| Census | 2500k | 13 | gender | 5000 | 1988 | 1500 |

## 4 STREAMING ALGORITHMS

Our sliding window algorithm uses a framework introduced by Braverman et al. (2020); Woodruff et al. (2023). This framework reduces clustering on sliding windows to an online coreset problem, via a standard merge-and-reduce method, and the detailed algorithm is listed in Algorithm 3. This algorithm maintains a coreset for the sliding window, and its guarantee is summarized in Theorem 4.1. The key idea is that every window can be viewed as a suffix of the input stream at some time step $t$, and if we build online coreset for all elements in the entire $1, \ldots, t$ time steps in the *reverse* order of time steps, then the prefix of the online coreset (which is the key query that the coreset provides) precisely gives a coreset for the sliding window.

---

**Algorithm 3** Sliding window coreset algorithm based on online assignment-preserving coreset

---

**procedure** MERGEANDREDUCE($P$)

    $t \leftarrow 0, \ell \leftarrow \lceil \lg m \rceil$

    initialize $\ell + 1$ point sets $B_0, B_1, \ldots B_\ell \leftarrow \varnothing$

    **while** input stream is not empty **do**

        $t \leftarrow t + 1$

        $p \leftarrow$ the next element in input stream, $\text{time}(p_i) \leftarrow -i$

        ▷online coresets is defined on the reverse stream

        let $j$ be the smallest index with $B_j = \varnothing$, set $j \leftarrow m$ if such index does not exist

        $B_j \leftarrow \text{ONLINECORESET}\left(\{p\} \cup B_0 \cup \ldots \cup B_{j-1}, \frac{\varepsilon}{2\log m}, \frac{\delta}{4m^2}\right)$

        clear the block $B_i$, i.e., $B_i \leftarrow \varnothing$, for all $i \in \{0, 1, \ldots, j-1\}$

        the coreset of the current window (i.e. the $t - m + 1, t - m + 2, \ldots t$-th element in input stream) is $\{p \in B_0 \cup B_1 \cup \ldots \cup B_\ell : \text{time}(p) \in [-t, -t + m)\}$

---

**Theorem 4.1.** *There exists an algorithm that takes as input $P \subseteq [\Delta]^d$ presented as a point stream, $0 < \varepsilon < 1, z \geq 1$, integer $k \geq 1$ and window size $m \geq 1$, computes a weighted subset $S$ of the sliding window after each point insertion, such that there exists an $\varepsilon$-coreset $S'$ of the sliding window for $(k, z)$-CLUSTERING with assignment constraints, that has $S' = S$ and $\forall p \in S, w_S(p) \in (1 \pm \varepsilon) w_{S'}(p)$. The algorithm uses space $\text{poly}(\varepsilon^{-1} kd \log(m\Delta))$, and succeeds with high probability.*

*Proof.* Consider Algorithm 3. Observe that for each $i \in \{0, 1, \ldots, \ell\}$, $B_i$ is an assignment-preserving online $((1 + \frac{\varepsilon}{2\log m})^i - 1)$-coreset for a sequence of $2^i$ consecutive points and the sequences of all $B_i$ ($i \in \{0, 1, \ldots, \ell\}$) are disjoint. Every time we set $B_i \leftarrow \text{ONLINECORESET}(\{p\} \cup B_0 \cup \ldots \cup B_{j-1}, \frac{\varepsilon}{2\log m}, \frac{\delta}{2m})$, $B_0, B_1, \ldots, B_{i-1}$ must be non-empty. By induction, $|\{p\} \cup B_0 \cup \ldots \cup B_{j-1}| = 1 + \sum_{j=0}^{i-1} 2^j = 2^i$ and the distortion of $B_i$ is at most $(1 + \frac{\varepsilon}{2\log m})^i$.

Suppose the input stream is $\{p_i\}_{i \geq 1}$. For every time $t$, after we have inserted the newly arrived point $p_t$, $B_0 \cup B_1 \cup \ldots \cup B_\ell$ is an assignment-preserving online $\varepsilon$-coreset of $\{p_t, p_{t-1}, \ldots, p_{t-2^\ell + 1}\}$ (sorted by time steps). By the definition of online coreset, $\{p \in B_0 \cup B_1 \cup \ldots \cup B_\ell : \text{time}(p) \in [-t, -t + m)\}$ is an assignment-preserving $\varepsilon$-coreset of $\{p_t, p_{t-1}, \ldots, p_{t-m+1}\}$, which is the current window. Since at most $2^\ell$ times of merge-and-reduce is involved in $B_0 \cup \ldots \cup B_{j-1}$, the failure probability for each time $t$ is at most $\delta$. This finishes the proof of Theorem 4.1. $\square$

*Proof of Theorem 1.2.* Our algorithm employs the following standard steps to obtain fair clustering from assignment-preserving coresets. Huang et al. (2019); Braverman et al. (2022) shows that, if dataset $P$ is partitioned with respect to the combinations of groups that each point belongs to, and

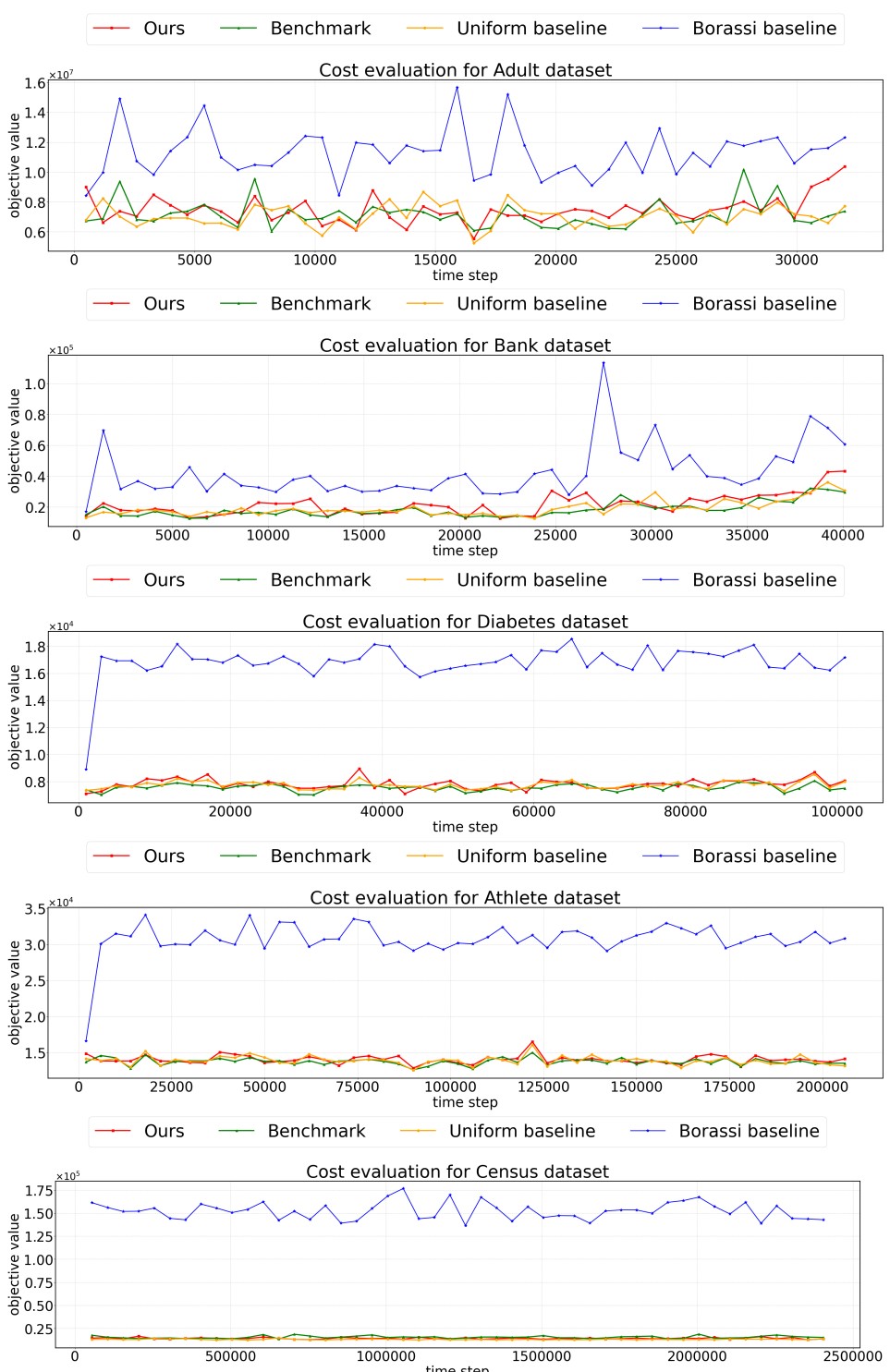

Fig. 1: Fair $k$-MEDIAN cost curves for all datasets.

suppose one takes the union of the assignment-preserving $\varepsilon$-coresets on each part, denoted as $S$, then the value of the optimal $(\alpha, \beta)$-fair clustering on $S$ is a $(1 + \varepsilon)$-approximation to that of $P$.

**Our algorithm.** Our algorithm follows similar steps. We apply Theorem 4.1 on the partition of $P$, then take the union to obtain some set $S'$. This $S'$ has size at most $L$ times that of the coreset

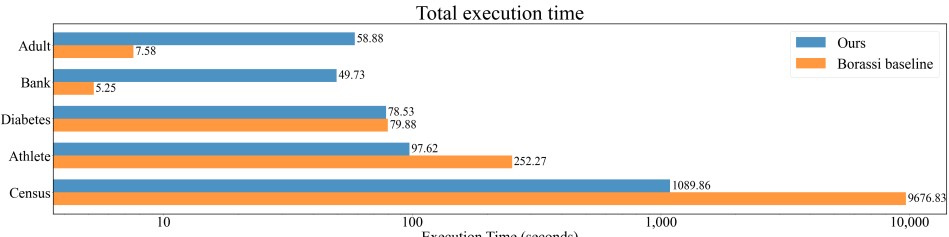

Fig. 2: Total running time for our algorithm and Borassi baseline over all sliding windows.

size in Theorem 4.1, and this dominates the space complexity. Finally, in the last step we find a $((1-\varepsilon)\alpha, (1+\varepsilon)\beta)$-fair clustering on $S'$ (which has a relaxed fairness constraint).

To see why this clustering has value at most $(1+\varepsilon)$ times the optimal $(\alpha, \beta)$-fair clustering on $P$, consider the optimal $(\alpha, \beta)$-fair solution on $S$ (the set defined in the first paragraph), then this solution is $(1+\varepsilon)$-approximate to that on $P$. Now, by the guarantee of Theorem 4.1, this same solution (with the same objective value) is a feasible $((1-\varepsilon)\alpha, (1-\varepsilon)\beta)$-fair clustering with respect to $S'$. Notice that what our algorithm finds is no worse than this solution on $S'$, and hence we conclude the clustering has cost at most $(1+\varepsilon)$ times the optimal solution. This finishes the proof. □

## 5 EXPERIMENTS

We implement our coresets and evaluate their performance for solving fair $k$-MEDIAN in sliding window. The evaluation requires a down-stream approximation algorithm for fair $k$-MEDIAN which is run on the coreset. We choose to use Fairtree (Backurs et al., 2019) in our experiments, which achieves competitive performance in practice.

**Baselines.** We employ three baselines. Two of the baselines are based on coresets, called "Benchmark" and "Uniform", which construct the coreset in an alternatively way than ours but still apply Fairtree as the down-stream approximation algorithm. Specifically, in "Benchmark", we construct a considerably large coreset from the sliding window (see Table 1 for the coreset sizes), and it serves as a benchmark for the accuracy; and in "Uniform", we use uniform sampling to build a coreset of the same size as ours, as a natural heuristic. The last baseline is a previous sliding-window algorithm designed for clustering without fairness constraints (Borassi et al., 2020), and we call it "Borassi".

**Datasets.** We evaluate the algorithms on 5 real datasets: Adult (Becker & Kohavi, 1996), Bank (Moro S & P, 2014), Diabetes (Kahn), Athlete (Barshan & Altun, 2010), and Census (Meek et al., 2001), which have also been used in various previous studies on (fair) clustering (Bera et al., 2019a; Chierichetti et al., 2017; Schmidt et al., 2018; Huang et al., 2019). For each dataset, we extract numerical features to construct a vector in $\mathbb{R}^d$ for each record, and we select a binary sensitive attribute. We set the window size $\approx \frac{n}{100}$. We list the detailed parameters of the datasets in Table 1.

**Experiment setup.** We choose $k = 10$ in all experiments. When implementing our coreset, we directly specify a target coreset size instead of using the worst-case bound as we established in previous sections. Due to variations in dataset sizes and corresponding window sizes, we assigned different coreset sizes for each dataset.

We set this target size 150 for both Adult and Bank, 300 for Diabetes, 750 for Athlete, and 1500 for Census. All the experiments are run on a MacBook Air 15.3 with an Apple M3 chip (8 cores, 2.22 GHz), 16GB RAM, and macOS 14.4.1 (23E224).

**Experiment results.** We run all algorithms on the five datasets, and we depict the cost curves in Figure 1 over the time steps. These curves show that our algorithm obtains a comparable cost to Benchmark and Uniform, while using lower space than Benchmark and achieving lower variance than Uniform (which is as expected, since the variance of uniform sampling can be unbounded even without fairness constraints). Finally, compared with Borassi, our algorithm performs better in accuracy, and as can be seen from Figure 2, we also achieve a better running time on larger datasets.

ACKNOWLEDGEMENTS

Research of Shaofeng Jiang was supported in part by a national key R&D program of China No. 2021YFA1000900 and a startup fund from Peking University. Samson Zhou is supported in part by NSF CCF-2335411. The work was conducted in part while Samson Zhou was visiting the Simons Institute for the Theory of Computing as part of the Sublinear Algorithms program. We thank the anonymous reviewers for insightful comments.

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

## A    LOWER BOUNDS

In this section, we show that fair clustering cannot be performed in the sliding window model in sublinear space, up to any finite multiplicative error.

We first define the one-way two-party augmented indexing communication problem. Formally, in the $\mathsf{AugInd}_n$ problem, Alice receives a vector $v \in \{0, 1\}^n$ and Bob receives an index $i \in [n]$, as well as the values of $v_1, \ldots, v_{i-1}$. The goal is for Alice to send some message $\Pi$ to Bob, so that Bob may compute $v_i$ with probability at least $\frac{2}{3}$. We recall the following communication complexity lower bounds for the augmented indexing problem.

**Theorem A.1.** *(Kremer et al., 1999) Any protocol that solves $\mathsf{AugInd}_n$ with probability at least $\frac{2}{3}$ requires $\Omega(n)$ bits of communication.*

We now show that the augmented indexing communication problem is hard even if it is promised that half of the entries of Alice's input vector are nonzero. Formally, in the $\mathsf{AugInd}_{2n}^n$ problem, Alice receives a vector $v \in \{0, 1\}^{2n}$, which has exactly $n$ nonzeros, and Bob receives an index $i \in [2n]$, as well as the values of $v_1, \ldots, v_{i-1}$. The goal is again for Alice to send some message $\Pi$ to Bob, so that Bob may compute $v_i$ with probability at least $\frac{2}{3}$.

**Corollary A.2.** *Any protocol that solves* $\mathsf{AugInd}_{2n}^n$ *with probability at least* $\frac{2}{3}$ *requires* $\Omega(n)$ *bits of communication.*

*Proof.* Suppose there exists a protocol $\Pi$ that solves $\mathsf{AugInd}_{2n}^n$ with probability at least $\frac{2}{3}$. Given an instance of $\mathsf{AugInd}_n$, let $s$ be the number of nonzero entries in Alice's input vector $v \in \{0, 1\}^n$. Then Alice can create a vector $u \in \{0, 1\}^{2n}$ with exactly $n$ nonzeros by appending to $v$ a total of $n - s$ coordinates that are 1, followed by $s$ coordinates that are 0. Since the number of nonzero entries in $v$ is $s$ and Alice appends $n - s$ nonzero entries afterwards, then it follows that $u$ has exactly $n$ nonzero coordinates. Alice and Bob can then run the protocol $\Pi$ for $\mathsf{AugInd}_{2n}^n$ on $u$ to find $u_i$, where $i \in [n]$ is the input to $\mathsf{AugInd}_n$. By construction, we have $u_i = v_i$, and thus the protocol solves $\mathsf{AugInd}_n$ with probability at least $\frac{2}{3}$, due to the correctness of $\Pi$. Hence, $\Pi$ requires $\Omega(n)$ bits of communication by Theorem A.1. $\qquad\square$

We now prove our lower bound for fair clustering in the sliding window model.

**Theorem 1.1.** *Any algorithm for fair $(k, z)$-clustering in the sliding window model that either achieves any multiplicative approximation or additive $\frac{\Delta}{2} - 1$ error, with probability at least $\frac{2}{3}$, must use $\Omega(n)$ space.*

*Proof.* Let $v \in \{0, 1\}^{2n}$ with exactly $n$ nonzero entries and $i \in [2n]$ be an input to $\mathsf{AugInd}_{2n}^n$. Alice creates an instance of fair $(k, z)$-clustering on $\mathbb{R}$, with $k = 2$. There are two groups $C_1, C_2$. The fairness constraint is that every cluster should contain at least $0.5$-fraction of points from $C_2$. For each $i \in [2n]$, Alice adds a point $x_i$ at $v_i \cdot \Delta$, all points in sequence $x$ (including those points which are defined later) belong to $C_1$. There are exactly $n$ points at the origin and exactly $n$ points at $\Delta$. Alice then creates a stream $S$ by inserting the points $x_1, \ldots, x_{2n}$ in order.

Let $\Pi$ be any algorithm for fair $(k, z)$-clustering in the sliding window model, with window size $4n$. Alice then runs $\Pi$ on $S$ and passes the state of the algorithm to Bob, who has $v_1, \ldots, v_{i-1}$ and thus also $x_1, \ldots, x_{i-1}$. Bob sets $x_{2n+j} = x_j$ for $j \in [i - 1]$ and inserts the points into the stream. Then Bob sets $x_{2n+i} = 0$ and inserts the point into the stream. Finally, Bob inserts $n$ points that are located at the origin and belong to $C_2$, another $n$ points that are located at $\Delta$ and belong to $C_2$. By construction, the active points in the window are precisely the $2n$ points from $C_1$, i.e. $x_{i+1}, \ldots, x_{2n+i}$ and $2n$ points from $C_2$.

Since all points are located at $\{0, \Delta\}$, the optimal center set is $\{0, \Delta\}$. The number of points from $C_1$ that are located at the origin equals to $|\{j \in [i + 1, 2n + i] : x_j = 0\}| = n + I(x_i = 1)$.

- If $x_i = 0$, then both location $0$ and $\Delta$ contains $n$ points from $C_1$ and $n$ points from $C_2$. The optimal solution is to assign every point $p$ to the center at the same location as $p$, which requires zero cost.

- If $x_i = 1$, then location $0$ contains $n + 1$ points from $C_1$ and $n$ points from $C_2$. In the optimal solution, either one of the points from $C_1$ and located at $0$ is assigned to center $\Delta$, or one of the points from $C_2$ and located at $\Delta$ is assigned to center $0$. For either case, the cost of optimal solution is no less than $\Delta$.

Thus if $\Pi$ achieves either any multiplicative approximation or additive error $\frac{\Delta}{2} - 1$ with probability at least $\frac{2}{3}$, then $\Pi$ can distinguish between these cases, thereby solving $\mathsf{AugInd}_{2n}^n$. Hence by Corollary A.2, $\Pi$ uses $\Omega(n)$ bits of space. $\qquad\square$

We can slightly modify the proof of Theorem 1.1 to obtain a lower bound of arbitrary additive violation $\delta$.

**Theorem A.3.** *Any algorithm that, with probability at least $2/3$, either achieves any multiplicative approximation or additive $\Delta/2 - 1$ error, even using $\delta$ additive violation in the fairness constraints must use $\Omega(n/\delta)$ space.*

*Proof.* Using the construction in the proof of Theorem 1.1, we can obtain a point sequence $A$ of length $\lfloor \frac{n}{\delta+1} \rfloor$, such that any algorithm for fair $(k, z)$-clustering in the sliding window that either

achieves any multiplicative approximation or additive $\Delta/2 - 1$ error on instance $A$ with probability at least $2/3$, must use $\Omega(n/\delta)$ space.

Next we construct sequence $B$ by repeating each point in $A$ for $\delta + 1$ times, i.e. $B = (A_1, A_1, \ldots, A_1, A_2, \ldots, A_2, A_3, \ldots, A_{\lfloor \frac{n}{\delta+1} \rfloor})$. In the proof of Theorem 1.1, we reduce $\text{AugInd}_{2n}^n$ to fair clustering in the sliding window by running the fair clustering algorithm on $A$ to distinguish whether window $(x_{i+1}, x_{i+2}, \ldots, x_{2n+i-1}, 0)$ has $n$ zeros or $n + 1$ zeros ($x$ is defined in the proof). Now suppose we have a fair clustering in the sliding window algorithm $\mathcal{A}$ which outputs a solution with at most $\delta$ additive violation and at most $\Delta/2 - 1$ additive error (to the optimal solution with no violation). Similar to the proof of Theorem 1.1, we can also distinguish whether window $(x_{i+1}, x_{i+2}, \ldots, x_{2n+i-1}, 0)$ has $n$ zeros or $n + 1$ zeros by running $\mathcal{A}$ on $B$, which completes the reduction from $\text{AugInd}_{2\lfloor \frac{n}{\delta+1} \rfloor}^{\lfloor \frac{n}{\delta+1} \rfloor}$ to fair clustering in the sliding window with additive violation. This implies $\mathcal{A}$ must use $\Omega(n/\delta)$ space. □

# B    MEYERSON SKETCH

We provide a brief overview of the Meyerson sketch (Meyerson, 2001) and its key properties relevant to our work. The Meyerson sketch achieves a bicriteria $(C_1, C_2)$-approximation for $(k, z)$-clustering on a data stream consisting of points $x_1, \ldots, x_n \in [\Delta]^d$, where $C_1 = 2^{z+7}$ and $C_2 = \mathcal{O}\left(2^{2z} \log n \log \Delta\right)$, i.e., it provides a $C_1$-approximation while using at most $C_2 k$ centers. An important feature of the Meyerson sketch that we shall utilize is that upon the arrival of each point $x_i$, the algorithm permanently assigns $x_i$ to one of the $C_2 k$ centers, even if there is subsequently a center closer to $x_i$ that is opened. Moreover, the clustering cost at the end of the stream is determined based on the center to which $x_i$ was assigned at time $i$.

To simplify the discussion, we describe the Meyerson sketch for the case where $z = 1$, noting that the intuition extends naturally to other values of $z$. The Meyerson sketch operates using a guess-and-double strategy, where it begins by estimating the optimal clustering cost. Based on this estimated cost, it randomly converts each point $x_i$ into a center, with probability proportional to the distance of the point $x_i$ from the existing centers at time $i$. If the algorithm opens too many centers, it concludes that the estimated optimal clustering cost was too low and doubles the value of the guess.

# C    PROOF OF LEMMA 3.3

**Lemma 3.3.** *Consider* $q \in [\Delta]^d$, $r > 0$, $\varepsilon, \delta \in (0,1), k, z \geq 1$, *and weighted set* $P = (p_1, p_2, \ldots, p_n) \subseteq [\Delta]^d \cap \text{ring}(q, r/2, r)$ *listed in the increasing order of time steps. Let* $N := w(P)$ *and* $S$ *be the output of* RINGCORESET$(P, \varepsilon, \delta)$ *(in Algorithm 2). We have* $E[|S|] = \text{poly}(2^z k \varepsilon^{-1} \log(N \Delta \varepsilon^{-1} \delta^{-1}))$, *and with probability at least* $1 - \delta$, *there exists a weighted set* $S'$ *such that*

- $S' = S$,
- *For every* $p \in S$, $w_{S'}(p) \in (1 \pm \varepsilon) w_S(p)$,
- *For every center set* $C \subseteq \mathbb{R}^d, |C| \leq k$ *and every assignment constraint* $\Gamma$ *with respect to* $P, C$, *we have* $|\text{cost}(S', C, \Gamma) - \text{cost}(P, C, \Gamma)| \leq \frac{\varepsilon}{4}(\text{cost}(P, C, \Gamma) + \text{cost}(S', C, \Gamma) + Nr^z)$.

To prove Lemma 3.3, we start with showing the following Lemma C.1 which has the online coreset error guarantee only for one center set $C$ and assignment constraint $\Gamma$, and only holds when all centers are close to $q$. The condition of Lemma C.1 is guaranteed by Lemma C.3, which reduces infinitely many pairs of $(C, \Gamma)$ with unbounded $\text{diam}(C)$ to finite pairs with $\text{diam}(C) = \mathcal{O}(z/\varepsilon) \cdot \text{diam}(P)$. We will conclude Lemma 3.3 by combining Lemma C.1 and Lemma C.3.

Define $R_{\text{far}} := 10zr/\varepsilon$ and $C^{\text{far}} := \{c \in C : d(q, c) > R_{\text{far}}\}$. Lemma C.1 assumes the point set $P$ lies in $B(q, R_{\text{far}})$.

**Lemma C.1.** *For every* $r > 0, \varepsilon, \delta \in (0,1), d, k, z \geq 1, q \in [\Delta]^d$, *weighted set* $P \subseteq [\Delta]^d \cap \text{ring}(q, r/2, r)$, *center set* $C \subseteq B(q, R_{\text{far}}), |C| \leq k$, *and assignment constraint* $\Gamma$ *with respect to* $P, C$, *let* $N = w(P)$ *and* $S$ *be the output of* RINGCORESET$(P)$. *We have* $E[|S|] = \text{poly}(2^z dk \varepsilon^{-1} \log(N \Delta \varepsilon^{-1} \delta^{-1}))$, *and with probability at least* $1 - \delta'$ *(let* $\delta'$ *follows the definition*

in Algorithm 2), there exists a weighted set $S'$ with $|\text{cost}(S', C, \Gamma) - \text{cost}(P, C, \Gamma)| \leq \varepsilon N r^z$, such that $S' = S$ and for every $p \in S$, $w_{S'}(p) \in (1 \pm \varepsilon) w_S(p)$.

Pick any $\frac{\varepsilon}{12z} r$-net of ball $B(q, R_{\text{far}})$, denoted by $\mathcal{C}_q$. In Lemma C.3, we reduce arbitrary center sets to subsets of $\mathcal{C}_q$. For assignment constraints, we discretize every value $\Gamma(c)$ into $H := \{i \cdot \frac{N}{t} : i = 0, 1, \ldots, t\}$, where $t := \lceil k^2 (10z/\varepsilon + 1)^z \rceil$.

Define
$$\mathcal{F} = \{(C, \Gamma) : C \subseteq \mathcal{C}_q, |C| \leq k, \forall c \in C, \Gamma(c) \in H, \Gamma(C) = w(P)\}.$$
We have $|\mathcal{F}| \leq \mathcal{O}(1) \cdot k^{\mathcal{O}(kd)} (z/\varepsilon)^{\mathcal{O}(kzd)}$.

We introduce the following generalized triangle inequality for the proof of Lemma C.3.

**Lemma C.2.** *(Braverman et al., 2022) For every $d, z \geq 1, \delta \in (0, 1], a, b, c \in \mathbb{R}^d$, the following inequality holds,*
$$d(a, b)^z \leq (1 + t)^{z-1} d(a, c)^z + (1 + t^{-1})^{z-1} d(b, c)^z.$$

**Lemma C.3.** *For every weighted set $P \subseteq \text{ring}(q, r/2, r)$, $k$-center set $C \subset \mathbb{R}^d$ and assignment constraint $\Gamma$ with respect to $P, C$, let $N := w(P)$, there exists $k$-point center set and assignment constraint $(C', \Gamma') \in \mathcal{F}$ such that*
$$\text{cost}(P, C, \Gamma) \in \text{cost}(P, C', \Gamma') \pm \frac{\varepsilon}{10} N r^z \pm \frac{\varepsilon}{4} \text{cost}(P, C, \Gamma) + \sum_{c \in C^{\text{far}}} \Gamma(c) d(c, q)^z.$$

*Proof.* First, we move every center $c$ outside $B(q, R_{\text{far}})$ (i.e. $c \in C^{\text{far}}$) to $q$, and move every center $c$ inside $B(q, R_{\text{far}})$ (i.e. $c \in C \backslash C^{\text{far}}$) to its nearest neighbor in $\mathcal{C}_q$. Suppose $c$ is moving to $\mathcal{C}_q(c)$, for every center $c$. Let $\tilde{C} := \mathcal{C}_q(C) = \{\mathcal{C}_q(c) : c \in C\}$. Let $\tilde{\Gamma}$ be the assignment constraint after movements, i.e. $\tilde{\Gamma}(c) = \sum_{c' \in C : \mathcal{C}_q(c') = c} \Gamma(c')$.

For every assignment function $\sigma$ w.r.t $P, C$ and assignment function $\tilde{\sigma}$ w.r.t $P, \tilde{C}$, $\sigma \sim \Gamma, \tilde{\sigma} \sim \tilde{\Gamma}$, we call $(\sigma, \tilde{\sigma})$ is corresponding if $\tilde{\sigma}(x, q) = \sum_{c' \in C^{\text{far}}} \sigma(x, c')$ and $\tilde{\sigma}(x, c) = \sigma(x, c)$ for every $x \in P, c \in C \backslash C^{\text{far}}$. In order to bound the difference between $\text{cost}(P, C, \Gamma)$ and $\text{cost}(P, \tilde{C}, \tilde{\Gamma})$, we consider the cost difference of every corresponding pair $(\sigma, \tilde{\sigma})$, i.e.

$$\sum_{x \in P} \left( \sum_{c \in C} \sigma(x, c) d(x, c)^z - \sum_{c \in \tilde{C}} \tilde{\sigma}(x, c) d(x, c)^z \right) = \sum_{c \in C} \sum_{x \in P} \sigma(x, c)(d(x, c)^z - d(x, \mathcal{C}_q(c))^z).$$

Define $\varepsilon' := (1 + \varepsilon/6)^{1/(z-1)} - 1 = \Theta(\varepsilon/z)$. For every center $c \in C^{\text{far}}$ and point $x \in P$, $c$ is moved to $q$. By Lemma C.2, we have

$$\begin{aligned}
\sigma(x, c)(d(x, c)^z - d(x, q)^z) &\leq \sigma(x, c)((1 + \varepsilon')^{z-1} d(c, q)^z + (1 + \varepsilon'^{-1})^{z-1} d(x, q)^z) \\
&\leq (1 + \varepsilon')^{z-1} \sigma(x, c) \left( d(c, q)^z + \varepsilon'^{-z+1} \left( \frac{\varepsilon}{10z} d(c, q) \right)^z \right) \\
&\leq (1 + \varepsilon/6) \sigma(x, c) \left( 1 + \frac{\varepsilon}{10z} \right) d(c, q)^z \\
&\leq (1 + \varepsilon/5) \sigma(x, c) d(c, q)^z
\end{aligned}$$

From the opposite direction, $\sigma(x, c)(d(x, c)^z - d(x, q)^z)$ is lower bounded by

$$\begin{aligned}
\sigma(x, c)(d(x, c)^z - d(x, q)^z) &\geq \sigma(x, c)((d(c, q) - d(x, q))^z - d(x, q)^z) \\
&\geq \sigma(x, c) \left( (d(c, q) - (\varepsilon/10z) d(c, q))^z - ((\varepsilon/10z) d(c, q))^z \right) \\
&\geq ((1 - (\varepsilon/10z))^z - (\varepsilon/10z)^z) \sigma(x, c) d(c, q)^z \\
&\geq (1 - \varepsilon/5) \sigma(x, c) d(c, q)^z
\end{aligned}$$

For every center $c \in C \backslash C^{\text{far}}$ and point $x \in P$, when $d(x, c') > d(x, c)$, moving $c$ to $c' = \mathcal{C}_q(c)$ increases the cost by

$$\sigma(x, c)(d(x, c')^z - d(x, c)^z) \leq \sigma(x, c)(((1 + \varepsilon')^{z-1} - 1) d(x, c)^z + (1 + \varepsilon'^{-1})^{z-1} d(c, c')^z)$$

$$\leq \sigma(x,c)\left(\varepsilon d(x,c)^z/6 + (1+\varepsilon/6)\varepsilon'^{-z+1}(\varepsilon r/12z)^z\right)$$
$$\leq \sigma(x,c)\left(\frac{\varepsilon}{6}d(x,c)^z + \frac{\varepsilon}{12}r^z\right).$$

When $d(x,c) > d(x,c')$, by the same argument we have,

$$\sigma(x,c)(d(x,c)^z - d(x,c')^z) \leq \sigma(x,c)(((1+\varepsilon')^{z-1}-1)d(x,c')^z + (1+\varepsilon'^{-1})^{z-1}d(c,c')^z)$$
$$\leq \sigma(x,c)\left(\frac{\varepsilon}{6}d(x,c)^z + \frac{\varepsilon}{12}r^z\right).$$

In summary,

$$\sum_{c\in C}\sum_{x\in P}\sigma(x,c)(d(x,c)^z - d(x,\mathcal{C}_q(c))^z)$$

$$\in \left(1 \pm \frac{\varepsilon}{6}\right)\sum_{c\in C^{\text{far}}}\sum_{x\in P}\sigma(x,c)d(c,q)^z \pm \frac{\varepsilon}{4}\sum_{c\in C\setminus C^{\text{far}}}\sum_{x\in P}\sigma(x,c)d(x,c)^z \pm \frac{\varepsilon}{12}Nr^z$$

$$\in \sum_{c\in C^{\text{far}}}\Gamma(c)d(c,q)^z \pm \frac{\varepsilon}{6}\sum_{c\in C^{\text{far}}}\sum_{x\in P}\sigma(x,c)(d(x,c) \pm d(x,q))^z$$
$$\pm \frac{\varepsilon}{4}\sum_{c\in C\setminus C^{\text{far}}}\sum_{x\in P}\sigma(x,c)d(x,c)^z \pm \frac{\varepsilon}{12}Nr^z$$

$$\in \sum_{c\in C^{\text{far}}}\Gamma(c)d(c,q)^z \pm \frac{\varepsilon}{6}\sum_{c\in C^{\text{far}}}\sum_{x\in P}(1 \pm (\varepsilon/10z))\sigma(x,c)d(x,c)^z$$
$$\pm \frac{\varepsilon}{4}\sum_{c\in C\setminus C^{\text{far}}}\sum_{x\in P}\sigma(x,c)d(x,c)^z \pm \frac{\varepsilon}{12}Nr^z$$

$$\in \sum_{c\in C^{\text{far}}}\Gamma(c)d(c,q)^z \pm \frac{\varepsilon}{4}\text{cost}(P,C,\Gamma) \pm \frac{\varepsilon}{12}Nr^z.$$

Since the bound holds for every corresponding $\sigma, \sigma'$, let $\sigma^*$ be the optimal assignment function of $\text{cost}(P,C,\Gamma)$ and $\tilde{\sigma}$ be some corresponding assignment function of $\sigma^*$, we have

$$\text{cost}(P,C,\Gamma) = \text{cost}^{\sigma^*}(P,C,\Gamma)$$
$$\geq \text{cost}^{\tilde{\sigma}}(P,\tilde{C},\tilde{\Gamma}) + \sum_{c\in C^{\text{far}}}\Gamma(c)d(c,q)^z - \frac{\varepsilon}{4}\text{cost}(P,C,\Gamma) - \frac{\varepsilon}{12}Nr^z$$
$$\geq \text{cost}(P,\tilde{C},\tilde{\Gamma}) + \sum_{c\in C^{\text{far}}}\Gamma(c)d(c,q)^z - \frac{\varepsilon}{4}\text{cost}(P,C,\Gamma) - \frac{\varepsilon}{12}Nr^z. \qquad (1)$$

On the opposite side, let $\tilde{\sigma}^*$ be the optimal assignment function of $\text{cost}(P,\tilde{C},\tilde{\Gamma})$ and $\sigma$ be some corresponding assignment function of $\tilde{\sigma}^*$, we have

$$\text{cost}(P,\tilde{C},\tilde{\Gamma}) = \text{cost}^{\tilde{\sigma}^*}(P,\tilde{C},\tilde{\Gamma})$$
$$\geq \text{cost}^{\sigma}(P,C,\Gamma) - \sum_{c\in C^{\text{far}}}\Gamma(c)d(c,q)^z - \frac{\varepsilon}{4}\text{cost}(P,C,\Gamma) - \frac{\varepsilon}{12}Nr^z$$
$$\geq \text{cost}(P,C,\Gamma) - \sum_{c\in C^{\text{far}}}\Gamma(c)d(c,q)^z - \frac{\varepsilon}{4}\text{cost}(P,C,\Gamma) - \frac{\varepsilon}{12}Nr^z. \qquad (2)$$

Combining (1) and (2), we have

$$\left|\text{cost}(P,C,\Gamma) - \text{cost}(P,\tilde{C},\tilde{\Gamma}) - \sum_{c\in C^{\text{far}}}\Gamma(c)d(c,q)^z\right| \leq \frac{\varepsilon}{4}\text{cost}(P,C,\Gamma) + \frac{\varepsilon}{12}Nr^z. \qquad (3)$$

Next, we find $\tilde{\Gamma}' : \tilde{C} \to H$ such that $\tilde{\Gamma}'(\tilde{C}) = N$ and for every $c \in \tilde{C}$, $|\tilde{\Gamma}(c) - \tilde{\Gamma}'(c)| < 1/t$, which always exists. On the other hand, there also exists assignment function $\sigma' \sim \Gamma'$ such that for every

$c \in \tilde{C}, \sum_{x \in P} |\sigma(x,c) - \sigma'(x,c)| < 1/t$. The difference of cost between $\sigma$ and $\sigma'$ is bounded by

$$
\begin{aligned}
|\operatorname{cost}(P, \tilde{C}, \tilde{\Gamma}) - \operatorname{cost}(P, \tilde{C}, \tilde{\Gamma}')| &\leq \left| \sum_{c \in \tilde{C}} \sum_{x \in P} (\sigma(x,c) - \sigma'(x,c)) d(x,c)^z \right| \\
&\leq \left| \sum_{c \in \tilde{C}} \sum_{x \in P} (\sigma(x,c) - \sigma'(x,c)) (\operatorname{diam}(\mathcal{C}_q) + 2r)^z \right| \\
&\leq (10z/\varepsilon + 1)^z \frac{k}{t} r^z \\
&\leq \frac{\varepsilon}{8} r^z.
\end{aligned}
\tag{4}
$$

Now we are able to give a error bound between $\operatorname{cost}(P, C, \Gamma)$ and $\operatorname{cost}(P, \tilde{C}, \tilde{\Gamma}')$ by summing up (3) and (4),

$$
\begin{aligned}
& \left| \operatorname{cost}(P, C, \Gamma) - \operatorname{cost}(P, \tilde{C}, \tilde{\Gamma}') - \sum_{c \in C^{\text{far}}} \Gamma(c) d(c,q)^z \right| \\
& \leq \left| \operatorname{cost}(P, C, \Gamma) - \operatorname{cost}(P, \tilde{C}, \tilde{\Gamma}) - \sum_{c \in C^{\text{far}}} \Gamma(c) d(c,q)^z \right| + |\operatorname{cost}(P, \tilde{C}, \tilde{\Gamma}) - \operatorname{cost}(P, \tilde{C}, \tilde{\Gamma}')| \\
& \leq \left( \frac{\varepsilon}{12} N + \frac{\varepsilon}{8} \right) r^z + \frac{\varepsilon}{4} \operatorname{cost}(P, C, \Gamma) \\
& \leq \frac{\varepsilon}{10} N r^z + \frac{\varepsilon}{4} \operatorname{cost}(P, C, \Gamma).
\end{aligned}
$$

We conclude the proof by picking $C' = \tilde{C}, \Gamma' = \tilde{\Gamma}'$.

$\square$

*Proof of Lemma 3.3.* By union bound, with probability at least $1 - \delta$, the statement of Lemma C.1 holds for every $\Gamma \in \mathcal{F}$.

For every $(C, \Gamma)$, let $(C', \Gamma') \in \mathcal{F}$ be the center set and assignment constraint stated in Lemma C.3 with respect to $(C, \Gamma)$. We have

$$
\begin{aligned}
|\operatorname{cost}(S, C, \Gamma) - \operatorname{cost}(P, C, \Gamma)| \\
\leq \quad & |\operatorname{cost}(S, C', \Gamma') - \operatorname{cost}(P, C', \Gamma')| + |(\operatorname{cost}(S, C, \Gamma) - \operatorname{cost}(S, C', \Gamma')) \\
& - (\operatorname{cost}(P, C, \Gamma) - \operatorname{cost}(P, C', \Gamma'))| \\
\leq \quad & |\operatorname{cost}(S, C', \Gamma') - \operatorname{cost}(P, C', \Gamma')| + \frac{\varepsilon}{5} N r^z + \frac{\varepsilon}{4} (\operatorname{cost}(S, C, \Gamma) + \operatorname{cost}(P, C, \Gamma)) \\
\leq \quad & \frac{\varepsilon}{4} (N r^z + \operatorname{cost}(S, C, \Gamma) + \operatorname{cost}(P, C, \Gamma)).
\end{aligned}
$$

Since the size of $S$ is independent of $C$ and $\Gamma$, the upper bound of $E[|S|]$ is the same as the result in Lemma C.1. This finishes the proof of Lemma 3.3. $\square$

## C.1 PROOF OF LEMMA C.1

**Lemma C.1.** *For every $r > 0, \varepsilon, \delta \in (0,1), d, k, z \geq 1, q \in [\Delta]^d$, weighted set $P \subseteq [\Delta]^d \cap \operatorname{ring}(q, r/2, r)$, center set $C \subseteq B(q, R_{\text{far}}), |C| \leq k$, and assignment constraint $\Gamma$ with respect to $P, C$, let $N = w(P)$ and $S$ be the output of RINGCORESET($P$). We have $E[|S|] = \operatorname{poly}(2^z d k \varepsilon^{-1} \log(N \Delta \varepsilon^{-1} \delta^{-1}))$, and with probability at least $1 - \delta'$ (let $\delta'$ follows the definition in Algorithm 2), there exists a weighted set $S'$ with $|\operatorname{cost}(S', C, \Gamma) - \operatorname{cost}(P, C, \Gamma)| \leq \varepsilon N r^z$, such that $S' = S$ and for every $p \in S$, $w_{S'}(p) \in (1 \pm \varepsilon) w_S(p)$.*

Some parts of the proof of this lemma are inspired by Cohen-Addad & Li (2019), but compared to Cohen-Addad & Li (2019), our proof needs to handle additional difficulties, such as dealing with weighted input points and non-uniform sampling probabilities.

As $w(S)$ may not equal to $w(P)$, $\text{cost}(S, C, \Gamma)$ is not well-defined since $\Gamma$ is not an assignment constraint with respect to $S$. Before discussing the cost of $(S, C, \Gamma)$, we need to generalize the definition of the cost function to the case that $w(S) \neq \Gamma(C)$. Then we prove Lemma C.1 by showing that both $|\text{cost}(P, C, \Gamma) - \text{cost}'(S, C, \Gamma)|$ and $|\text{cost}(S', C, \Gamma) - \text{cost}'(S, C, \Gamma)|$ are small.

**Definition C.4.** *For weighted point set $P \subseteq [\Delta]^d$ and a center set $C \subseteq \mathbb{R}^d$, a partial assignment function with respect to $P$ and $C$ is a function $\sigma : P \times C \to \mathbb{R}_{\geq 0}$. For a partial assignment function $\sigma$ with respect to $P, C$ and an assignment constraint $\Gamma$ with respect to $P', C$, where $P'$ is some arbitrary weighted point set ($P'$ is only used in $\Gamma(C) = w(P')$), we say $\sigma$ is partially consistent with $\Gamma$ if the following property holds, denoted by $\sigma \sim' \Gamma$.*

- *For every $p \in P$, $\sum_{c \in C} \sigma(p, c) \leq w_P(p)$.*

- *For every $c \in C$, $\sum_{p \in P} \sigma(p, c) \leq \Gamma(c)$.*

- *If $w(P) \geq \Gamma(C)$, then $\sigma$ satisfies that for every $c \in C$, $\sum_{p \in P} \sigma(p, c) = \Gamma(c)$.*

- *If $w(P) \leq \Gamma(C)$, then $\sigma$ satisfies that for every $p \in P$, $\sum_{c \in C} \sigma(p, c) = w_P(p)$.*

A partial assignment function can be viewed as a maximum flow of the following flow network.

- For every $p \in P$, there is an edge from source node to $p$ with capacity $w_P(p)$.

- For every $c \in C$, there is an edge from $c$ to sink node with capacity $\Gamma(c)$.

- For every $p \in P, c \in C$, there is an edge from $p$ to $c$ with infinite capacity.

Now we can define the cost of $S$ with respect to the constraint of $\Gamma$.

$$\text{cost}'(S, C, \Gamma) = \min_{\sigma : S \times C \to \mathbb{R}_{\geq 0}, \sigma \sim' \Gamma} \sum_{p \in P} \sum_{c \in C} \sigma(p, c) \, \text{dist}(p, c)^z.$$

**Lemma C.5.** *With probability at least $1 - \delta/2$, $|\text{cost}'(S, C, \Gamma) - \text{cost}(P, C, \Gamma)| \leq \varepsilon N r^z / 2$.*

*Proof.* The proof is postponed to Appendix C.2. $\qquad\square$

**Lemma C.6.** *With probability at least $1 - \delta/2$, there exists a weighted point set $S'$ with $|\text{cost}'(S, C, \Gamma) - \text{cost}(S', C, \Gamma)| \leq \varepsilon N r^z / 2$, such that $S' = S$ and for every $p \in S$, $w_{S'}(p) \in (1 \pm \varepsilon) w_S(p)$.*

*Proof.* The proof is postponed to Appendix C.3. $\qquad\square$

**Lemma C.7.** *Let $T$, $\text{prob}$ follow the definition in Algorithm 2, $\sum_{i=1}^{n} \text{prob}_i \leq T \ln N + 1$.*

*Proof.* Notice that

$$\text{sum}_n \prod_{i=2}^{n} \left( 1 - \frac{w_P(p_i)}{\text{sum}_n} \right) = w_P(p_1),$$

$$\prod_{i=2}^{n} \left( 1 - \frac{w_P(p_i)}{\text{sum}_n} \right)^{-1} = \frac{\text{sum}_n}{w_P(p_1)},$$

$$\sum_{i=2}^{n} \frac{w_P(p_i)}{\text{sum}_i} \leq -\sum_{i=2}^{n} \ln \left( 1 - \frac{w_P(p_i)}{\text{sum}_n} \right) = \ln \frac{\text{sum}_n}{w_P(p_1)} \leq \ln N$$

So we have

$$\sum_{i=1}^{n} \text{prob}_i = 1 + \sum_{i=2}^{n} \min \left( \frac{T \cdot w_P(p)}{\text{sum}_i}, 1 \right) \leq T \ln N + 1.$$

$\qquad\square$

*Proof of Lemma C.1.* By applying a union bound to the results of Lemma C.5 and Lemma C.6, we can show that with probability at least $1 - \delta$, there exists a weighted point set $S'$ with $|\cost(P, C, \Gamma) - \cost(S', C, \Gamma)| \leq \varepsilon N r^z$, such that $S' = S$ and for every $p \in S$, $w_{S'}(p) \in (1 \pm \varepsilon)w_S(p)$.

By Lemma C.7, $E[|S|] = \sum_{i=1}^{n} \prob_i \leq T \ln N + 1$. $\qquad\qquad\qquad\qquad\qquad\qquad\qquad\qquad\qquad\qquad\qquad \square$

## C.2   PROOF OF LEMMA C.5

To prove $|\cost'(S, C, \Gamma) - \cost(P, C, \Gamma)| \leq \varepsilon N r^z / 2$, we show that given an optimal assignment function $\sigma$ of $\cost(P, C, \Gamma)$, we can always construct a partial assignment function with cost no more than $\cost(P, C, \Gamma) + \varepsilon N r^z / 2$, vice versa.

In the following discussion, let $n, T, p_1, \ldots, p_n, \text{sum}_1, \ldots, \text{sum}_n, \prob_1, \ldots, \prob_n$ follow the definition in Algorithm 2 and let $S$ be the output of RINGCORESET$(P, \varepsilon, \delta)$. For every $1 \leq i \leq n$, let indicator variable $X_i$ denotes $I(p_i \in S)$.

We first prove a concentration inequality used in the following proofs.

**Lemma C.8.** *For every $0 < \varepsilon, \delta < 1, W > 0, \alpha_1, \alpha_2, \ldots, \alpha_n$ such that $\forall 1 \leq i \leq n, 0 \leq \alpha_i \leq W \cdot w_P(p_i)$, as long as $T = \Omega(\varepsilon^{-2} \log(\delta^{-1}\varepsilon^{-1})(\log\log(NW))^2)$, we have*

$$\Pr\left[\left|\sum_{i=1}^{n} \alpha_i \prob_i^{-1} X_i - \sum_{i=1}^{n} \alpha_i\right| > \varepsilon \sum_{i=1}^{n} \alpha_i + \varepsilon NW\right] \leq \delta.$$

*Proof.* Here we assume all points $p_i$ satisfy $\prob_i < 1$, otherwise we always have $\alpha_i \prob_i^{-1} X_i = \alpha_i$ and we can eliminate these points from $P$ without changing $\left|\sum_{i=1}^{n} \alpha_i \prob_i^{-1} X_i - \sum_{i=1}^{n} \alpha_i\right|$. Hence we always have $\prob_i = \frac{T \cdot w_P(p_i)}{\text{sum}_i}$. As a result, for every $1 \leq i \leq n$, we have

$$\alpha_i \prob_i^{-1} \leq W \cdot w_P(p_i) \cdot \frac{\text{sum}_i}{T \cdot w_P(p_i)} \leq \frac{NW}{T}.$$

Let $\gamma = 1 + \frac{\varepsilon}{8 \ln N}$, we divide $P$ into $L = \mathcal{O}\left(\varepsilon^{-1} \log N \log(\varepsilon^{-1}nNW/T)\right)$ groups $G_{j_{\min}}, G_{j_{\min}+1}, \ldots, G_{j_{\max}} \subseteq [n]$, where $j_{\min} = \lceil\log_\gamma(\varepsilon/n)\rceil$, $j_{\max} = \lceil\log_\gamma(NW/T)\rceil$. Group $G_j$ contains the indices of all points $p_i$ such that $\alpha_i \prob_i^{-1} \in (\gamma^{j-1}, \gamma^j]$. In particular, $G_{j_{\min}}$ contains all points $p_i$ such that $\alpha_i \leq \gamma^{j_{\min}}$.

For every $j_{\min} < j \leq j_{\max}$, we have

$$\left|\sum_{i \in G_j} \alpha_i \prob_i^{-1} X_i - \sum_{i \in G_j} \alpha_i\right| \leq \gamma^j \left|\sum_{i \in G_j} X_i - \sum_{i \in G_j} \prob_i\right| + (\gamma - 1)\gamma^{j-1} \sum_{i \in G_j}(X_i + \prob_i) \quad (5)$$

For $j = j_{\min}$ we have $\sum_{i \in G_{j_{\min}}} \alpha_i \prob_i^{-1} \leq \varepsilon$.

For the second term of (5), it is sufficient to bound $\sum_{i \in G_j} \prob_i$ since by Chernoff bound we have

$$\Pr\left[\sum_{i=1}^{n} X_i > \max\left(2\sum_{i=1}^{n} \prob_i, 4\ln(2/\delta)\right)\right] \leq \delta/2.$$

By Lemma C.7, with probability at least $1 - \delta/2$, we have

$$\sum_{j=j_{\min}}^{j_{\max}} (\gamma - 1)\gamma^{j-1} \sum_{i \in G_j}(\prob_i + X_i) \leq (\gamma - 1)\gamma^{j_{\max}-1} \sum_{i=1}^{n}(\prob_i + X_i)$$

$$\leq (\gamma - 1)\gamma^{j_{\max}-1}\left(2\sum_{i=1}^{n} \prob_i + 4\ln(1/\delta)\right)$$

$$\leq (\gamma - 1)\frac{NW}{T}(2T \ln N + 4\ln(1/\delta) + 2)$$

$$\leq \frac{\varepsilon NW}{2}. \tag{6}$$

Next consider the first term of (5). Let $\mu_j = \sum_{i \in G_j} \mathrm{prob}_i$, $A_j = \sum_{i \in G_j} \alpha_i$, notice that

$$\mu_j = \sum_{i \in G_j} \mathrm{prob}_i \leq \sum_{i \in G_j} \frac{\alpha_i}{\gamma^{j-1}} = \frac{A_j}{\gamma^{j-1}}.$$

By Chernoff bound, let $t = \max(\varepsilon A_j, 3\varepsilon^{-1}\gamma^{j+1}\ln(2L/\delta))$, we have

$$\Pr\left[\left|\sum_{i \in G_j} X_i - \mu_j\right| \geq t\gamma^{-j}\right] \leq 2\exp\left(-\frac{t^2}{3\gamma^{2j}\mu_j}\right)$$

$$\leq 2\exp\left(-\frac{t^2}{3\gamma^{j+1}A_j}\right)$$

$$\leq \frac{\delta}{2L}.$$

By a union bound, with probability at least $1 - \delta/2$, we have

$$\sum_{j=j_{\min}}^{j_{\max}} \gamma^j \left|\sum_{i \in G_j} X_i - \sum_{i \in G_j} \mathrm{prob}_i\right| \leq \varepsilon \sum_{j=j_{\min}}^{j_{\max}} A_j + 3\varepsilon^{-1}\sum_{j=j_{\min}}^{j_{\max}} (1+\varepsilon_0)^{j+1}\ln(2L/\delta)$$

$$\leq \varepsilon\sum_{i=1}^{n} \alpha_i + 3\varepsilon^{-2}\gamma^{j_{\max}+1}\ln(2L/\delta)$$

$$\leq \varepsilon\sum_{i=1}^{n} \alpha_i + 3\varepsilon^{-2}\gamma^2\ln(2L/\delta)\max_{i \in [n]}\alpha_i\mathrm{prob}_i^{-1}$$

$$\leq \varepsilon\sum_{i=1}^{n} \alpha_i + \frac{3\varepsilon^{-2}\gamma^2\ln(2L/\delta)}{T}NW$$

$$\leq \varepsilon\sum_{i=1}^{n} \alpha_i + \frac{\varepsilon}{2}NW. \tag{7}$$

Combining (6) and (7) by a union bound, we have

$$\Pr\left[\left|\sum_{i=1}^{n}\alpha_i\mathrm{prob}_i^{-1}X_i - \sum_{i=1}^{n}\alpha_i\right| > \varepsilon\sum_{i=1}^{n}\alpha_i + \varepsilon NW\right] \leq \delta.$$

$\square$

Define $R = (20z/\varepsilon + 2)r \geq \mathrm{diam}(C) + \mathrm{diam}(P)$. For every pair $c \in C, p \in P$, $\mathrm{dist}(p, c)$ is bounded by $R$. Let $\varepsilon_0 = (\varepsilon/22z)^{-z-1} \leq (20z/\varepsilon + 2)^{-z}\varepsilon$. In the following proofs, we show these errors are bounded by $\varepsilon_0 NR^z$ and that implies $\mathrm{err} \leq \varepsilon_0 NR^z \leq \varepsilon Nr^z$.

**Lemma C.9.** $\Pr[\mathrm{cost}'(S, C, \Gamma) \leq \mathrm{cost}(P, C, \Gamma) + \varepsilon Nr^z/2] \geq 1 - \delta/4$.

*Proof.* To imply $\mathrm{cost}'(S, C, \Gamma) \leq \mathrm{cost}(P, C, \Gamma) + \varepsilon Nr^z/2$, we show that given an optimal assignment function $\sigma : P \times C \to \mathbb{R}_{\geq 0}$ of $\mathrm{cost}(P, C, \Gamma)$ there always exists a partial assignment function $\sigma' : S \times C \to \mathbb{R}_{\geq 0}$ for $\mathrm{cost}'(S, C, \Gamma)$ such that $\sum_{p \in S}\sum_{c \in C}\mathrm{dist}(p, c)^z\sigma'(p, c) \leq \sum_{p \in P}\sum_{c \in C}\mathrm{dist}(p, c)^z\sigma(p, c) + \varepsilon Nr^z/2$.

A natural way for constructing $\sigma'$ is to assign $\frac{\sigma(p,c)}{w_P(p)}w_S(p)$ units of weight from $p$ to $c$, denoted by assignment $\tau$. Let the cost of this assignment be

$$\mathrm{cost}_\tau := \sum_{p \in S}\sum_{c \in C}\frac{\sigma(p,c)}{w_P(p)}w_S(p) \cdot d(p,c)^z.$$

However, $\tau$ may not be consistent with $\Gamma$. Since $S$ is a random point set, some centers may receive more than $\Gamma(c)$ units of weight, denoted by center set $C^+$, and some may receive less than $\Gamma(c)$ units denoted by center set $C^-$. We transform $\tau$ into an assignment consistent with $\Gamma$ by collecting the surplus weights from $C^+$ and send them to $C^-$ so that $\sigma'$ is partially consistent with $\Gamma$. Re-routing $w$ weight started at point $p$ from $c$ to $c'$ costs at most $w(\text{dist}(p,c')^z - \text{dist}(p,c)^z) \le w \cdot R^z$. Hence this step costs at most

$$R^z \sum_{c \in C} \left| \Gamma(c) - \sum_{p \in S} \frac{\sigma(p_i, c)}{w_P(p_i)} w_S(p_i) \right| = R^z \sum_{c \in C} \left| \Gamma(c) - \sum_{i=1}^n \sigma(p_i, c) \text{prob}_i^{-1} X_i \right|$$

Taking $\varepsilon' = \frac{\varepsilon_0}{8k}, \delta' = \frac{\delta}{8k}, W = 1, \forall 1 \le i \le n, \alpha_i = \sigma(p_i, c)$ in Lemma C.8, we have for every $c \in C$,

$$\Pr \left[ \left| \Gamma(c) - \sum_{i=1}^n \sigma(p_i, c) \text{prob}_i^{-1} X_i \right| > \frac{\varepsilon_0}{8k} (\Gamma(c) + N) \right] \le \frac{\delta}{4k}$$

By taking union bound over every $c \in C$, with probability at least $1 - \delta/8$, we have

$$|\text{cost}_\tau - \text{cost}'(S, C, \Gamma)| \le R^z \sum_{c \in C} \left| \Gamma(c) - \sum_{i=1}^n \sigma(p_i, c) \text{prob}_i^{-1} X_i \right| \le \frac{\varepsilon_0}{4} N R^z.$$

Next, we show that the cost of $\tau$ is close to the real cost of $P$.

$$|\text{cost}_\tau - \text{cost}(P, C, \Gamma)| \le \left| \sum_{c \in C} \sum_{i=1}^n \text{dist}(p_i, c)^z \left( \frac{\sigma(p_i, c)}{w_P(p_i)} \cdot w_S(p_i) - \sigma(p_i, c) \right) \right|$$

Taking $\varepsilon' = \frac{\varepsilon_0}{8k}, \delta' = \frac{\delta}{8k}, W = R^z, \forall 1 \le i \le n, \alpha_i = \text{dist}(p_i, c)\sigma(p_i, c)$ in Lemma C.8, we have for every $c \in C$,

$$\Pr \left[ \left| \sum_{i=1}^n \text{dist}(p_i, c)^z \left( \frac{\sigma(p_i, c)}{w_P(p_i)} \cdot w_S(p_i) - \sigma(p_i, c) \right) \right| > \frac{\varepsilon_0}{8k} (\text{cost}(P, C, \Gamma) + N R^z) \right] \le \frac{\delta}{8k}$$

By taking union bound over every $c \in C$, with probability at least $1 - \delta/8$, we have

$$\sum_{c \in C} \left| \sum_{i=1}^n \text{dist}(p_i, c)^z \left( \frac{\sigma(p_i, c)}{w_P(p_i)} \cdot w_S(p_i) - \sigma(p_i, c) \right) \right| \le \frac{\varepsilon_0}{4} N R^z.$$

In summary, with probability at least $1 - \delta/4$, we have

$$\text{cost}'(S, C, \Gamma) - \text{cost}(P, C, \Gamma) \le (\text{cost}'(S, C, \Gamma) - \text{cost}_\tau) + (\text{cost}_\tau - \text{cost}(P, C, \Gamma))$$
$$\le \frac{\varepsilon_0}{2} N R^z \le \frac{\varepsilon}{2} N r^z.$$

$\square$

Since $S$ is a random set and contains much fewer points than $P$, constructing an assignment function with respect to $P, C$ from a partial assignment function of $\text{cost}'(S, C, \Gamma)$ is difficult. Hence we first compare $\text{cost}(P, C, \Gamma)$ to the expectation of $\text{cost}'(S, C, \Gamma)$ and then show the concentration bound of $\text{cost}'(S, C, \Gamma)$.

**Lemma C.10.** $\text{cost}(P, C, \Gamma) \le E[\text{cost}'(S, C, \Gamma)] + \varepsilon N r^z / 4$.

*Proof.* Let $\sigma : P \times C \to \mathbb{R}_{\ge 0}$ be an assignment function with respect to $P, C$. Let $\sigma^*$ be a random partial assignment function that denotes the optimal assignment function for $\text{cost}'(S, C, \Gamma)$.

Since $\sigma^* \sim' \Gamma$, we have $\forall c \in C, \sum_{p \in P} E[\sigma^*(p, c)] \le \Gamma(c)$ and $\forall p \in P, \sum_{c \in C} E[\sigma^*(p, c)] \le w_P(p)$. We can construct $\sigma$ by initializing $\sigma(p, c)$ as $E[\sigma^*(p, c)]$, then arbitrarily assigning the

remaining weights of data points to the remaining capacities of centers. This assigning step costs at most

$$
\begin{aligned}
R^z \left( N - \sum_{p \in P} \sum_{c \in C} E[\sigma^*(p,c)] \right) &= R^z \left( N - E \left[ \sum_{p \in P} \sum_{c \in C} \sigma^*(p,c) \right] \right) \\
&= R^z E[N - \min\{w(S), N\}] \qquad (\sigma^* \sim \Gamma) \\
&= R^z E[\max\{N - w(S), 0\}] \\
&\leq R^z E \left[ \left| \sum_{i=1}^{n} w_P(p_i) \mathrm{prob}_i^{-1} X_i - \sum_{i=1}^{n} w_P(p_i) \right| \right]
\end{aligned}
$$

Taking $\varepsilon' = \frac{\varepsilon_0}{8}, \delta' = \frac{\varepsilon_0}{Nw(P)}, W = 1, \forall 1 \leq i \leq n, \alpha_i = w_P(p_i)$ in Lemma C.8, we have

$$
\Pr \left[ \left| \sum_{i=1}^{n} w_P(p_i) \mathrm{prob}_i^{-1} X_i - \sum_{i=1}^{n} w_P(p_i) \right| > \frac{\varepsilon_0}{8} N \right] \leq \frac{\varepsilon_0}{Nw(P)},
$$

which implies

$$
R^z E \left[ \left| \sum_{i=1}^{n} w_P(p_i) \mathrm{prob}_i^{-1} X_i - \sum_{i=1}^{n} w_P(p_i) \right| \right] \leq R^z \left( \frac{\varepsilon_0 N}{8} + \frac{\varepsilon_0 n \cdot \mathrm{sum}_n}{Nw(P)} \right) \leq \frac{\varepsilon_0}{4} N R^z \leq \frac{\varepsilon}{4} N r^z.
$$

$\square$

We use a concentration inequality based on martingales (Chung & Lu, 2006) to show the concentration bound of $\mathrm{cost}'(S, C, \Gamma)$. For convenience, we use $x_{l...r}$ to represent $(x_l, x_{l+1}, \ldots, x_r)$ for every vector $\mathbf{x}$ of dimension $d$, $1 \leq l \leq r \leq d$. This notation is also used for function arguments, e.g. $f(x_{1...i})$ represents $f(x_1, x_2, \ldots, x_i)$.

**Lemma C.11.** *(Chung & Lu, 2006) Let $V = \mathcal{X}_1 \times \mathcal{X}_2 \times \ldots \times \mathcal{X}_n$ ($\mathcal{X}_1, \mathcal{X}_2, \ldots, \mathcal{X}_n \subseteq \mathbb{R}$) be a set of $n$-dimensional real vectors. Suppose there is a function $f : V \to \mathbb{R}$ and $n$ distributions $D_1, D_2, \ldots, D_n$ over $\mathcal{X}_1, \mathcal{X}_2, \ldots, \mathcal{X}_n$ respectively and $\mathcal{D}$ is the joint distribution over $V$, Define $f^{(i)}(z_1, z_2, \ldots, z_i) = E[f(\mathbf{x}) \mid x_1 = z_1, \ldots, x_i = z_i]$.*

*If $\sigma_1^2, \sigma_2^2, \ldots, \sigma_n^2$ satisfies for every $1 \leq i \leq n, z_1 \in \mathcal{X}_1, \ldots, z_{i-1} \in \mathcal{X}_{i-1}$,*

$$
\mathrm{Var}_{x \sim D_i}(f^{(i)}(z_1, z_2, \ldots z_{i-1}, x)) \leq \sigma_i^2,
$$

*and $M$ satisfies that for every $1 \leq i \leq n, \mathbf{z} \in V, z_i' \in \mathcal{X}_i$ such that $\Pr_{\mathbf{x} \sim \mathcal{D}}[\mathbf{x} = \mathbf{z}] > 0, \Pr_{\mathbf{x} \sim \mathcal{D}}[x_1 = z_1, \ldots, x_i = z_i', \ldots, x_n = z_n] > 0$, we have*

$$
|f(z_1, z_2, \ldots, z_n) - f(z_{1...i-1}, z_i', z_{i+1...n})| \leq M,
$$

*then for every $t > 0$,*

$$
\Pr_{\mathbf{x} \sim \mathcal{D}}[|f(\mathbf{x}) - E_{\mathbf{x} \sim \mathcal{D}}[f(\mathbf{x})]| > t] \leq 2 \exp \left( \frac{-t^2}{2(\sum_{i=1}^{n} \sigma_i^2 + Mt/3)} \right).
$$

**Lemma C.12.** $\Pr[|\mathrm{cost}'(S, C, \Gamma) - E[\mathrm{cost}'(S, C, \Gamma)]| \leq \varepsilon N r^z / 4] \geq 1 - \delta/8.$

*Proof.* Define function $f : \mathbb{R}_{\geq 0}^n \to \mathbb{R}_{\geq 0}$ as $f(x_1, x_2, \ldots, x_n) := \mathrm{cost}'(R, C, \Gamma)$ where $R := \{p_i : i \in [n], x_i > 0\}$ and $w_R(p_i) := x_i$.

Define random vector $\mathbf{x} = (x_1, x_2, \ldots, x_n)$ be the vector representation of $S$, i.e.

$$
x_i = \begin{cases} w_S(p_i) & p_i \in S \\ 0 & p_i \notin S \end{cases}.
$$

Let $\mathcal{D}$ be the distribution of the weights $\mathbf{x}$, where

$$
x_i = \begin{cases} w_S(p_i) & p_i \in S \\ 0 & p_i \notin S \end{cases},
$$

and $S$ is the output of RingCoreset$(P)$.

By the definition of $\mathbf{x}$, $x_1, x_2, \ldots, x_n$ are independent random variables. If $\text{prob}_i = 1$, then $x_i$ always equals to $w_P(p_i)$. Otherwise, $x_i$ has a two-point distribution

$$
x_i = \begin{cases} \frac{\text{sum}_i}{T} & \text{w.p. } \text{prob}_i \\ 0 & \text{w.p. } 1 - \text{prob}_i \end{cases}.
$$

We assume $\text{prob}_i < 1$ in the following discussion since the conclusions obviously hold when $\text{prob}_i = 1$.

Suppose every $x_1, x_2, \ldots, x_n$ is fixed except $x_i$. In $f(x_1, x_2, \ldots, x_n)$, adding $x_i$ by 1 corresponds to adding $w_S(p_i)$ by 1, which increases $\text{cost}'(S, C, \Gamma)$ by at most $R^z$ (recall that every partial assignment function can be viewed as a maximum flow of certain flow network). Hence the difference of $f(x_1, \ldots, x_{i-1}, 0, \ldots, x_n)$ and $f(x_1, \ldots, x_{i-1}, \frac{\text{sum}_i}{T}, \ldots, x_n)$ is at most $R^z \frac{\text{sum}_i}{T} \leq R^z \frac{N}{T}$.

Define $f^{(i)}(z_1, z_2, \ldots, z_i) = E[f(\mathbf{x}) \mid x_1 = z_1, \ldots, x_i = z_i]$. To give an upper bound of $\text{Var}(f^{(i)}(z_{1 \ldots i-1}, x_i))$ ($\mathbf{z} \in \mathbb{R}^n, \Pr_{\mathbf{x} \sim \mathcal{D}}[\mathbf{x} = \mathbf{z}] > 0$), we first consider the difference between $F_0 := f^{(i)}(z_{1 \ldots i-1}, 0)$ and $F_1 := f^{(i)}(z_{1 \ldots i-1}, \text{sum}_i/T)$. By the arguments above, we have

$$
|F_0 - F_1| \leq \sum_{z_{i+1}, \ldots, z_n \in \mathbb{R}} \Pr[x_{i+1 \ldots n} = z_{i+1 \ldots n}] \cdot \left| f(z_{1 \ldots i-1}, 0, z_{i+1 \ldots n}) - f\left(z_{1 \ldots i-1}, \frac{\text{sum}_i}{T}, z_{i+1 \ldots n}\right) \right|
$$

$$
\leq \frac{R^z \text{sum}_i}{T}.
$$

Then we have

$$
\begin{aligned}
\text{Var}(f^{(i)}(z_{1 \ldots i-1}, x_i)) &= \text{prob}_i F_1^2 + (1 - \text{prob}_i) F_0^2 - (\text{prob}_i F_1 + (1 - \text{prob}_i) F_0)^2 \\
&= \text{prob}_i (1 - \text{prob}_i)(F_0 - F_1)^2 \\
&\leq \text{prob}_i (R^z \text{sum}_i/T)^2 \\
&\leq \frac{R^{2z} N}{T} w_P(p_i).
\end{aligned}
$$

Taking $M = R^z N/T$ and $\sigma_i^2 = \frac{R^{2z} N}{T} w_P(p_i)$ in Lemma C.11, we have

$$
\Pr\left[\text{cost}'(S, C, \Gamma) \leq E[\text{cost}'(S, C, \Gamma)] - \frac{\varepsilon_0}{4} N R^z\right]
$$

$$
\leq 2\exp\left(\frac{\varepsilon_0^2 N^2 R^{2z}}{32(\sum_{i=1}^n (R^{2z} N w_P(p_i)/T) + \varepsilon_0 M N R^z/3)}\right)
$$

$$
\leq 2\exp\left(\frac{\varepsilon_0^2 T}{32}\right)
$$

$$
\leq \delta/8.
$$

□

**Lemma C.5.** *With probability at least $1 - \delta/2$, $|\text{cost}'(S, C, \Gamma) - \text{cost}(P, C, \Gamma)| \leq \varepsilon N r^z/2$.*

*Proof.* By applying union bound to results of Lemma C.9, Lemma C.10 and Lemma C.12, we get $\Pr[|\text{cost}'(S, C, \Gamma) - \text{cost}(P, C, \Gamma)| \leq \varepsilon N r^z/2] \geq 1 - \delta/2$. This concludes the proof. □

### C.3 Proof of Lemma C.6

**Lemma C.6.** *With probability at least $1 - \delta/2$, there exists a weighted point set $S'$ with $|\text{cost}'(S, C, \Gamma) - \text{cost}(S', C, \Gamma)| \leq \varepsilon N r^z/2$, such that $S' = S$ and for every $p \in S$, $w_{S'}(p) \in (1 \pm \varepsilon) w_S(p)$.*

*Proof.* Taking $\varepsilon' = \frac{\varepsilon_0}{4}, \delta' = \delta/2, W = 1, \forall 1 \leq i \leq n, \alpha_i = w_P(p_i)$ in Lemma C.8, we have

$$
\Pr\left[\frac{w(S)}{N} \in 1 \pm \frac{\varepsilon_0}{4}\right] = \Pr\left[\sum_{p_i \in S} w_P(p_i) \text{prob}_i^{-1} X_i \in \left(1 \pm \frac{\varepsilon_0}{4}\right) \sum_{p_i \in P} w_P(p_i)\right]
$$

$$\geq 1 - \delta/2.$$

Let $\alpha = \frac{w(S)}{N}$. Conditioning on $\alpha \in 1 \pm \frac{\varepsilon_0}{4}$, we show that $|\text{cost}'(S, C, \Gamma) - \text{cost}(S', C, \Gamma)| \leq \varepsilon_0 N R^z / 2$ always holds, where $S'$ contains all elements in $S$ and the weights is defined by $w_{S'}(x) = \alpha^{-1} w_S(x)$ for every $x \in S$.

For every partial assignment function $\sigma$ with respect to $S, C$ and partially consistent with $\Gamma$, and assignment function $\sigma'$ with repsect to $S', C$ and consistent with $\Gamma$, we have $\sum_{x \in P} \sum_{c \in C} \sigma'(x, c) = N$ and $\sum_{x \in P} \sum_{c \in C} \sigma(x, c) = \min(w(P), N) = \min(\alpha, 1)N$. In the following construction, we use $\min(\alpha, 1)$ to normalize the weight of $\sigma$.

First we prove that $\text{cost}'(S, C, \Gamma) \leq \text{cost}(S', C, \Gamma) + \varepsilon N r^z / 2$. Let $\sigma'$ be the optimal assignment function in $\text{cost}(S', C, \Gamma)$. We define the partial assignment function $\sigma$ as $\sigma(p, c) = \min(\alpha, 1)\sigma'(p, c)$ for every $p \in P, c \in C$. We can verify that $\sigma$ is partially consistent with $\Gamma$.

$$\begin{aligned}
\text{cost}'(S, C, \Gamma) &\leq \sum_{p \in P} \sum_{c \in C} \sigma(p, c) \, \text{dist}(p, c)^z \\
&\leq \min(\alpha, 1) \, \text{cost}(S', C, \Gamma) \\
&\leq \text{cost}(S', C, \Gamma).
\end{aligned}$$

Next we prove that $\text{cost}(S', C, \Gamma) \leq \text{cost}'(S, C, \Gamma) + \varepsilon N r^z / 2$. Let $\sigma$ be the optimal partial assignment function in $\text{cost}'(S, C, \Gamma)$. We initialize the assignment function $\sigma'$ as $\sigma'(p, c) = \max(\alpha^{-1}, 1)\sigma(p, c)$. However, $\sigma'$ may not be consistent with $\Gamma$. In order to obtain an assignment function consistent with $\Gamma$, we apply the following modifications to $\sigma'$.

- Let $S'_+ = \{x \in S' : \sum_{c \in C} \sigma'(x, c) > w_{S'}(x)\}$, $S'_- = \{x \in S' : \sum_{c \in C} \sigma'(x, c) < w_{S'}(x)\}$. For every center $c \in C$, we re-assign weights from $\sigma'(x, c)$ ($x \in S'_+$) to $\sigma'(x', c)$ ($x' \in S'_-$). Repeating this process until $\sum_{c \in C} \sigma'(x, c) = w_{S'}(x)$ for every $x \in P$. This step costs at most $R^z \sum_{x \in S'} |w_{S'}(x) - \sum_{c \in C} \max(\alpha^{-1}, 1)\sigma(x, c)|$.

- Let $C_+ = \{c \in C : \sum_{x \in P} \sigma'(x, c) > \Gamma(c)\}$, $C_- = \{c \in C : \sum_{x \in P} \sigma'(x, c) < \Gamma(c)\}$. Notice that the previous step does not affect $\sum_{x \in P} \sigma'(x, c)$. For every point $x \in S'$, we re-assign weights from $\sigma'(x, c)$ ($c \in C_+$) to $\sigma'(x, c')$ ($c' \in C_-$). Repeating this process until $\sum_{x \in S'} \sigma'(x, c) = \Gamma(c)$ for every $c \in C$. This step costs at most $R^z \sum_{c \in C} |\Gamma(c) - \max(\alpha^{-1}, 1) \sum_{x \in S'} \sigma(x, c)|$.

After these modifications, the cost of $\sigma'$ is bounded by

$$\text{cost}^{\sigma'}(P, C, \Gamma) - \text{cost}'(P, C, \Gamma)$$
$$\leq \ R^z \sum_{x \in S'} \left| w_{S'}(x) - \max(\alpha^{-1}, 1) \sum_{c \in C} \sigma(x, c) \right| + R^z \sum_{c \in C} \left| \Gamma(c) - \max(\alpha^{-1}, 1) \sum_{x \in S'} \sigma(x, c) \right|$$

If $w(S) \geq N$ (i.e. $\alpha^{-1} \leq 1$), then $\sum_{x \in S'} \sigma(x, c) = \Gamma(c)$ holds for every center $c \in C$.

$$\begin{aligned}
\text{cost}^{\sigma'}(P, C, \Gamma) - \text{cost}'(P, C, \Gamma) \\
\leq \ & R^z \sum_{x \in S'} \left| w_{S'}(x) - \sum_{c \in C} \sigma(x, c) \right| + R^z \sum_{c \in C} \left| \Gamma(c) - \sum_{x \in S'} \sigma(x, c) \right| \\
= \ & R^z \sum_{x \in S'} \left| w_{S'}(x) - \sum_{c \in C} \sigma(x, c) \right| \\
\leq \ & R^z \sum_{x \in S'} (w_S(x) - w_{S'}(x)) + \left( w_S(x) - \sum_{c \in C} \sigma(x, c) \right) \\
\leq \ & 2R^z (w(S) - N) \leq \frac{\varepsilon_0}{2} N R^z
\end{aligned}$$

If $w(S) < N$ (i.e. $\alpha^{-1} > 1$), then $\sum_{c \in C} \sigma(x,c) = w_S(x)$ holds for every $x \in P$.

$\text{cost}^{\sigma'}(P,C,\Gamma) - \text{cost}'(P,C,\Gamma)$

$$
\begin{aligned}
&\leq\ R^z \sum_{x \in S'} \left| w_{S'}(x) - \alpha^{-1} \sum_{c \in C} \sigma(x,c) \right| + R^z \sum_{c \in C} \left| \Gamma(c) - \alpha^{-1} \sum_{x \in S'} \sigma(x,c) \right| \\
&=\ R^z \alpha^{-1} \sum_{x \in S'} \left( w_S(x) - \sum_{c \in C} \sigma(x,c) \right) + R^z \sum_{c \in C} \left| \Gamma(c) - \alpha^{-1} \sum_{x \in S'} \sigma(x,c) \right| \\
&=\ R^z \sum_{c \in C} \left| \Gamma(c) - \alpha^{-1} \sum_{x \in S'} \sigma(x,c) \right| \\
&=\ R^z \sum_{c \in C} (\alpha^{-1} - 1)\Gamma(c) + \alpha^{-1} \left( \Gamma(c) - \sum_{x \in S'} \sigma(x,c) \right) \\
&=\ 2R^z(\alpha^{-1} - 1)N \leq \frac{\varepsilon_0}{2} N R^z.
\end{aligned}
$$

Summing up together we get $\text{cost}(S',C,\Gamma) \leq \text{cost}'(S,C,\Gamma) + \varepsilon_0 N R^z/2$. This concludes the proof.

$\square$

