# OpenReview forum: "Fair Clustering in the Sliding Window Model"
_ICLR.cc/2025/Conference — ICLR 2025 Spotlight_

### Official Review · Reviewer_5dNf · 2024-10-23

**Soundness:** 3
**Presentation:** 3
**Contribution:** 3
**Rating:** 8
**Confidence:** 3

**Summary:**

This paper investigates the problem of fair clustering in the sliding window model. First, it presents the inapproximability of the problem. Then, it proposes a new coreset-based approximation algorithm for the problem by relaxing the fairness constraint. Finally, some experimental results are provided to show that the proposed algorithm is better than trivial uniform sampling.

**Strengths:**

S1. This paper addresses an open theoretical problem - whether the proportional fairness constraint and the sliding window model are compatible in sublinear space - by providing the inapproximability result.

S2. The proposed algorithm is the first known one with an approximation guarantee for fair clustering in the sliding window model.

S3. The theoretical results are thorough and solid.

S3. The paper is generally well-written and organized.

**Weaknesses:**

W1. Although the theoretical part of this paper is sound and solid, the experimental part is highly insufficient.
- At least, two additional types of baselines should be compared: (1) the algorithms for sliding-window clustering without fairness such as those in [Borassi et al., 2020; Epasto et al., 2022;  Woodruff et al., 2023], which indicates "the price of fairness", and the algorithms for offline and insert-only streaming with fairness such as those in [Chierichetti et al., 2017; Schmidt et al., 2018; Huang et al., 2019], which provides lower bounds of clustering costs and presents the challenge of the sliding-window model. Uniform sampling is a too-weak baseline for coreset-based clustering.
- How about the performance of the proposed algorithms on datasets with higher dimensions (e.g., [Census](https://archive.ics.uci.edu/dataset/116/us+census+data+1990) and synthetic data in about 10-100 dimensions)?
- Details about experimental setup and implementation (such as dataset preprocessing and code availability) are concise.
- The efficiency results (e.g., running time and coreset size) and scalability (e.g., performance w.r.t. k and d) are not provided. For example, the clustering cost, runtime, and coreset size w.r.t. different values of k and d can be presented in figures or tables.
- According to the theoretical results, it seems that the proposed algorithms can work for multiple $l>2$ attribute groups. However, the experiments only use a binary attribute for each dataset.

W2. Minor presentation problems. Please double-check the paper carefully. Just one example is presented here.
- In the abstract, "*we show that if the fairness constraint by a multiplicative $\varepsilon$ factor, ...*" missing "is relaxed"?

**Questions:**

See the questions listed in W1.

---

> ### Author Response · Authors · 2024-11-22
>
> > W1. Although the theoretical part of this paper is sound and solid, the experimental part is highly insufficient.
>
> > At least, two additional types of baselines should be compared: (1) the algorithms for sliding-window clustering without fairness such as those in [Borassi et al., 2020; Epasto et al., 2022; Woodruff et al., 2023], which indicates "the price of fairness", and the algorithms for offline and insert-only streaming with fairness such as those in [Chierichetti et al., 2017; Schmidt et al., 2018; Huang et al., 2019], which provides lower bounds of clustering costs and presents the challenge of the sliding-window model. Uniform sampling is a too-weak baseline for coreset-based clustering.
>
> Thanks for the suggestion. We agree that uniform sampling is not the best baseline for many datasets and thus we have conducted additional empirical evaluations, incorporating both larger datasets and new baselines. Specifically, we evaluated both (1) an algorithm for clustering in the sliding window model that does not consider fairness (Borassi et al. 2020) and (2) an algorithm for fair clustering in the offline setting that does not consider the sliding window model (Backurs et al., 2019). Our experiments demonstrate that our algorithm performs better than (Borassi et al. 2020) in accuracy (using similar time and space), and achieves a comparable objective value while using much lower space than (Backurs et al., 2019) which needs to store the whole sliding window.
>
> > How about the performance of the proposed algorithms on datasets with higher dimensions (e.g., Census and synthetic data in about 10-100 dimensions)?
>
> We have performed additional experiments on the Census dataset. Our results indicate that our algorithm achieves a consistent performance as with other datasets.
>
> > Details about experimental setup and implementation (such as dataset preprocessing and code availability) are concise.
>
> As we already mentioned in the submitted draft, the dataset is processed by extracting numerical features to construct a vector in $R^d$ for each data point. However, we also performed some standard normalization, which was not mentioned. In the next version, we will discuss more details of the experiments, and will make our source code for the experiments available (as a github repo).
>
> > The efficiency results (e.g., running time and coreset size) and scalability (e.g., performance w.r.t. k and d) are not provided. For example, the clustering cost, runtime, and coreset size w.r.t. different values of k and d can be presented in figures or tables.
>
> We have provided additional details on running time for the algorithms across the various datasets. The running time is reported for proper sliding window baselines, i.e., ours and the new Borassi baseline. Our algorithm has a comparable running time to Borassi while achieving a much better objective value.
>
> > According to the theoretical results, it seems that the proposed algorithms can work for multiple $l>2$ attribute groups. However, the experiments only use a binary attribute for each dataset.
>
> It is correct that our algorithm can work for multiple attribute groups. In fact, the way to implement this is very straightforward: one simply builds a coreset for each (non-empty) attribute group, and take the union. Hence, there is no real reason that we cannot build such coresets efficiently.
>
> However, the crucial limiting factor is the downstream approximation algorithms for fair clustering: the Fairtree algorithm by (Backurs et al., 2019) which we use only supports binary attribute gruops, and we are not aware of a (practically) efficient algorithm for multiple attribute groups; For instance, (Bera et al., 2019) can handle multiple attribute groups but it is based on LP-rounding which can be slow. This makes it difficult to evaluate the multiple attribute groups case.
>
> > W2. Minor presentation problems. Please double-check the paper carefully. Just one example is presented here.
>
> > In the abstract, "we show that if the fairness constraint by a multiplicative $\epsilon$ factor, ..." missing "is relaxed"?
>
> Thanks, we have taken additional passes over the paper, focusing on overall presentation.

---

> > ### Comment · Reviewer_5dNf · 2024-11-25
> >
> > Thank you for the clarifications. My concerns have been mostly addressed. I will raise my score.

---

### Official Review · Reviewer_RZvP · 2024-10-30

**Soundness:** 4
**Presentation:** 2
**Contribution:** 3
**Rating:** 8
**Confidence:** 2

**Summary:**

In the paper, the authors consider solving the problem of achieving (1+\epsilon) multiplicative approximate fair clustering in the fixed-size sliding window model. The author proved that there exists a sublinear space algorithm to solve the ((1-\epsilon)\alpha, (1+\epsilon)\beta) clustering problem. The core solution builds on top of online corset algorithms. At a high level, it first constructs a set of sketches to approximate the cluster, then decomposes the clusters into rings, and lastly union the uniform samples from the rings.

**Strengths:**

The paper studies an important problem with praticial impacts.

It not only presents novel theoretical insights but also implemented the algorithm. The evaluation showcased that the proposed algorithm consistently achieves better results compared to a uniform sampling baseline.

**Weaknesses:**

It would be nice to introduce a bit more on the (Augemented) Meyerson Sketches.

Please add more discussions and implications of multiplicative/additive approximation error on applications..

**Questions:**

See weakness.

---

> ### Author Response · Authors · 2024-11-22
>
> > It would be nice to introduce a bit more on the (Augemented) Meyerson Sketches.
>
> Thanks for the suggestion. We will add the following text describing the Meyerson sketch in the next version of the manuscript:
>
> We provide a brief overview of the Meyerson sketchand its key properties relevant to our work. The Meyerson sketch achieves a bicriteria $(C_1,C_2)$-approximation for $(k,z)$-clustering on a data stream consisting of points $x_1,\ldots,x_n\in [\Delta]^d$, where $C_1=2^{z+7}$ and $C_2=O(2^{2z}\log n\log \Delta)$, i.e., it provides a $C_1$-approximation while using at most $C_2k$ centers. An important feature of the Meyerson sketch that we shall utilize is that upon the arrival of each point $x_i$, the algorithm permanently assigns $x_i$ to one of the $C_2 k$ centers, even if there is subsequently a center closer to $x_i$ that is opened. Moreover, the clustering cost at the end of the stream is determined based on the center to which $x_i$ was assigned at time $i$.
>
> To simplify the discussion, we describe the Meyerson sketch for the case where $z=1$, noting that the intuition extends naturally to other values of $z$. The Meyerson sketch operates using a guess-and-double strategy, where it begins by estimating the optimal clustering cost. Based on this estimated cost, it randomly converts each point $x_i$ into a center, with probability proportional to the distance of the point $x_i$ from the existing centers at time $i$. If the algorithm opens too many centers, it concludes that the estimated optimal clustering cost was too low and doubles the value of the guess.
>
> > Also, I would like to know if the sketches used are mergeable [1]? If it is mergeable, then one may perhaps develope algortithms that merge smaller corsets to sovle the sliding window problem?
>
> > [1] Agarwal, Pankaj K., et al. "Mergeable summaries." ACM Transactions on Database Systems (TODS) 38.4 (2013): 1-28.
>
> Yes, the summaries in the provided reference [1] are mergeable, but they do not address the problem of clustering (or fairness), so while they can potentially be used to solve other problems in the sliding window model, they cannot be used in the context of fair clustering.

---

> > ### Comment · Reviewer_RZvP · 2024-11-25
> >
> > Thank you for the clarifications. Reviewing the feedback and responses from the other reviewer has given me a clearer understanding of your contributions. I have raised my score.

---

### Official Review · Reviewer_cQ32 · 2024-11-03

**Soundness:** 4
**Presentation:** 3
**Contribution:** 4
**Rating:** 8
**Confidence:** 3

**Summary:**

This paper studies the fair clustering problem within the sliding window model, where the goal is to maintain approximation for the most recdent data points in a stream with minimum space compelxity used. This paper first identifies a unique separation in clustering performance between the insertion-only and sliding window models, specifically for fair clustering, where no algorithm can achieve a bounded approximation ratio under strict fairness constraints without requiring linear space. To address this issue, ths paper then introduces an approximation algorithm that achieves a $(1 + \epsilon)$-approximation for fair clustering by allowing slight relaxation in fairness constraints, using sublinear space in $poly(k, \epsilon^{-1}, log n)$. Theoretical contributions include a strong lower bound proof, supported by communication complexity techniques, establishing the necessity of linear space for strict fairness in the sliding window model. Complementing these theoretical findings, this paper presents empirical results on real-world datasets, demonstrating that their approach significantly improves clustering cost and stability over a uniform sampling baseline, supporting the algorithm’s practical effectiveness in dynamic, fairness-sensitive applications.

**Strengths:**

1. Novelty and Insight: The paper identifies a crucial separation between fair clustering in different streaming models, offering new theoretical insights into the limitations of fair clustering in the sliding window model without any fairness violations.

2. Theoretical Contributions: The lower bound result is strong and well-supported by communication complexity arguments, which can have positive impact on the theoretical understanding of fair clustering.

3. Approximation Guarantee: The proposed approximation algorithm provides a $(1+\epsilon)$-approximation with sublinear space, which is a valuable contribution in the context of fair clustering and sliding windows.

4. Clear Presentation of Methods: The algorithms, particularly the coreset construction, are explained with clarity, and the theoretical proofs, while complex, are well-structured.

**Weaknesses:**

1. The space complexities used for the proposed method are dependent on the aspect ratio of the given clustering instances (an $O(\log\Delta)$ term, where $\Delta$ is the aspect ratio). Although, the aspect ratio can usually be assumed to be bounded by a polynomial function of the data size, it can be arbitrarily large in the worst case.

2. The empirical comparison primarily involves a uniform sampling baseline. A more comprehensive evaluation with other clustering or local search methods would strengthen the experimental section and provide a clearer picture of the proposed method’s effectiveness.

**Questions:**

1. What impact does the window size $W$ have on performance? Since sliding windows dynamically retain recent data, it would be insightful to see more analysis on how varying $W$ affects the clustering quality and computational efficiency of the proposed algorithm.

2. Is there any methods that can remove the dependence of aspect ratio on the space complexity? How to deal with the case if the aspect ratio of the given clustering instance is large. In previous work, it was pointed out that the log function of aspect ratio can be linearly dependent on the data size $n$ [1]. Additionally, the method in [1] provides an efficient way for reducing the aspect ratio of any arbitrary clustering instance to bounded $poly(n, d)$ in static setting.

[1] Draganov A, Saulpic D, Schwiegelshohn C. Settling Time vs. Accuracy Tradeoffs for Clustering Big Data[J]. Proceedings of the ACM on Management of Data, 2024, 2(3): 1-25.

**Details Of Ethics Concerns:**

Since this is mainly a theoretical result, there is no ethical issues that need further considerations.

---

> ### Author Response · Authors · 2024-11-22
>
> > The space complexities used for the proposed method are dependent on the aspect ratio of the given clustering instances (an $O(\log\Delta)$ term, where $\Delta$ is the aspect ratio). Although, the aspect ratio can usually be assumed to be bounded by a polynomial function of the data size, it can be arbitrarily large in the worst case.
>
> > Is there any methods that can remove the dependence of aspect ratio on the space complexity? How to deal with the case if the aspect ratio of the given clustering instance is large. In previous work, it was pointed out that the log function of aspect ratio can be linearly dependent on the data size $n$ [1]. Additionally, the method in [1] provides an efficient way for reducing the aspect ratio of any arbitrary clustering instance to bounded $poly(n,k)$ in static setting.
>
> We remark that the space complexity provably requires $\Omega(\log\Delta)$ bits of space. Informally, each coordinate of a data point must be encoded as both input and storage. However, since the aspect ratio is $\Delta$, then each coordinate requires $\Omega(\log\Delta)$ bits to represent. Moreover, the coordinates of some input point must be stored, because a center may be placed at that point, e.g., if the remaining points are also at that point. Thus intutiively, the algorithm must use $\Omega(\log\Delta)$ bits of space, and this can be formalized into a necesssary lower bound. Hence, there are provably no methods that can remove the dependence of aspect ratio on the space complexity. Note that this is not contradicted by [1], because their coreset construction bounds are in terms of number of points, not bits of space. Indeed, it is a standard assumption that $O(\log\Delta)=O(\log n)$, such as in [1], in which case our algorithms provide the same guarantees.
>
> [1] Draganov A, Saulpic D, Schwiegelshohn C. Settling Time vs. Accuracy Tradeoffs for Clustering Big Data[J]. Proceedings of the ACM on Management of Data, 2024, 2(3): 1-25.
>
> > The empirical comparison primarily involves a uniform sampling baseline. A more comprehensive evaluation with other clustering or local search methods would strengthen the experimental section and provide a clearer picture of the proposed method’s effectiveness.
>
> Thanks for the feedback. Unfortunately, local search is not compatible with either fairness or the sliding window model and we found it challenging to implement a meaningful analog. Nevertheless, we have performed additional experiments using larger datasets and new baselines. In particular, we evaluated both (1) an algorithm for clustering in the sliding window model that does not consider fairness (Borassi et al. 2020) and (2) an algorithm for fair clustering in the offline setting that does not consider the sliding window model (Backurs et al., 2019). Our experiments demonstrate that our algorithm performs better than (Borassi et al. 2020) in accuracy (using similar time and space), and achieves a comparable objective value while using much lower space than (Backurs et al., 2019) which needs to store the whole sliding window.
>
> > What impact does the window size $W$ have on performance? Since sliding windows dynamically retain recent data, it would be insightful to see more analysis on how varying $W$ affects the clustering quality and computational efficiency of the proposed algorithm.
>
> Choosing the correct window size $W$ is an interesting design question. If $W$ is too small, then potentially large amounts of useful information is discarded. Conversely, if $W$ is too large, then potentially large amounts of outdated information is included within the dataset, and the overall runtime and space complexity becomes larger. Thus it is not surprising that different companies choose to different values of $W$ in their data retention policies. We believe a comprehensive study on the sliding window value parameter across various tasks on various datasets would be of signficant interest to data analysts, but perhaps beyond the scope of this work.

---

> > ### Comment · Reviewer_cQ32 · 2024-11-25
> > **Response to the Rebuttal**
> >
> > Thanks for the clarification. All of my concerns have been fully addressed. In general, this is an interesting paper with solid theoretical analysis. I will maintain my initial evaluation.

---

### Official Review · Reviewer_hpoL · 2024-11-04

**Soundness:** 3
**Presentation:** 2
**Contribution:** 4
**Rating:** 6
**Confidence:** 3

**Summary:**

This paper studies the fair clustering problem in the sliding window model, where the goal is to maintain an approximation for clustering the most recent data points while satisfying proportional fairness constraints. Specifically, given a $k$-clustering instance in the form of a data stream, the goal of this paper is to achieve a $(1+\epsilon)$-approximation to the optimal fair clustering for the dataset defined by the sliding window with minimum space complexity. However, ensuring fairness under strict space constraints in a data stream poses significant challenges. To tackle the challenges in sliding window model, this paper proposes an online assignment-preserving coreset construction method. The proposed method first reduces clustering in the sliding window model to an online coreset construction via a standard merge-and-reduce technique. Then, this paper introduces an algorithm for assignment-preserving coreset construction that processes each window as a suffix of the input stream at a particular time $t$. Finally, by constructing the coreset in reverse order across all time steps from $1$ to $t$, the resulting prefix of this online coreset serves as a valid coreset for the sliding window, achieving $(1+\epsilon)$-approximation for the fair clustering problem in sliding window model. This paper shows that if the fairness constraint is allowed to be violated by a multiplicative factor, there exists a $(1+\epsilon)$-approximate sliding window algorithm that uses only $poly(k\epsilon^{−1}log n)$ space. Empirical evaluations on real-world datasets further validates the effectiveness of the proposed framework, complementing the theoretical results.

**Strengths:**

The strengths of this paper can be summarized as follows.

1. The theoretical results of the paper are solid.

2. This paper establishes lower bounds for the fair clustering in sliding window model.

3. The proposed method achieves near-optimal clustering performances with  $(1+\epsilon)$-approximation on clustering quality guarantees while the space complexity nearly matches the lower bound provided.

4. The proposed method uses sublinear space in sliding window model, which can be used for handling large, dynamic datasets that require efficient memory usage.

**Weaknesses:**

1. The proposed algorithm uses a multiplicative relaxation rather than an additive violation for fairness constraints, which is slightly different from previous fair clustering algorithms.

2. Although the theoretical results nearly match the lower bound, the space complexity still depends on the aspect ratio of the given clustering instances.

3. The paper lacks a sufficient number of comparison algorithms, making the experimental results less convincing. Additionally, the parameter choices and values for $\alpha$ and $\beta$ are not specified, limiting the reproducibility and clarity of the experimental parts.

**Questions:**

Q1: The proposed algorithm violates the fairness constraint by a multiplicative factor, which is slightly different from previous fair clustering algorithms with additive violations. Does multiplicative loss lead to better approximation ratios than previous algorithms with additive loss? What happens when only additive violation is allowed for group fairness constraints in the sliding window model, as achieving an approximation ratio of $1+\epsilon$ for group fair clustering is challenging.

Q2: As mentioned in the paper, the prefix property of online assignment-preserving coresets can tolerate a $1±\epsilon$ relative error in the weights. Does this property make the proposed algorithm easier to implement compared to the algorithm in [1]?

Q3: The comparison algorithms used this paper are limited, which makes the numerical experiments not convincing enough. The authors should add more sliding window algorithms, fair algorithms or heuristic algorithms as comparisons to make the experimental parts better.

Q4: How to determine the value of parameter $k$? Does different choices of $k$ influence the experimental results of the proposed algorithms? What are the choices of the values for $\alpha$ and $\beta$.

[1] Woodruff, David, Peilin Zhong, and Samson Zhou. "Near-Optimal k-Clustering in the Sliding Window Model." Advances in Neural Information Processing Systems 36 (2024).

---

> ### Author Response · Authors · 2024-11-22
>
> > Although the theoretical results nearly match the lower bound, the space complexity still depends on the aspect ratio of the given clustering instances.
>
> It can be shown that the space complexity necessarily depends on the aspect ratio of input clustering instances. In particular, each coordinate of a data point must be encoded as both input and storage. Since the aspect ratio is $\Delta$, then each coordinate requires $\Omega(\log\Delta)$ bits of space simply to represent. Moreover, it can be shown that the coordinates of some input point must be stored, because a center may be placed at that point, e.g., if the remaining points are also at that point. Hence, this notion can be formalized into showing that $\Omega(\log\Delta)$ bits of space is necessary.
>
> > Q1: … Does multiplicative loss lead to better approximation ratios than previous algorithms with additive loss? What happens when only additive violation is allowed for group fairness constraints in the sliding window model…
>
> Although certain previous works on fair clustering study algorithms with additive violations, e.g., [BCFN19], their focus is on polynomial runtime, whereas our focus in the sliding window model is on the space complexity of the algorithm. In particular, it can be shown that any algorithm that is permitted additive violation $\delta$ to the fairness constraint and achieves any multiplicative approximation or additive $\Delta/2-1$ error still requires $\Omega(W/\delta)$ space, which is prohibitively large. Therefore, this indicates that additive violation for fairness constraint is not enough for the sliding window model. We will add this proof formally to the next version of the manuscript.
>
> [BCFN19] Suman Kalyan Bera, Deeparnab Chakrabarty, Nicolas Flores, Maryam Negahbani: Fair Algorithms for Clustering. NeurIPS 2019: 4955-4966
>
> > Q2: As mentioned in the paper, the prefix property of online assignment-preserving coresets can tolerate a $1\pm\epsilon$ relative error in the weights. Does this property make the proposed algorithm easier to implement compared to the algorithm in [1]?
>
> Our notion of online assignment-preserving coresets is a strictly stronger notion than the online coreset of [1], as we also maintain a $(1+\varepsilon)$-approximation of the weights of the points in an assignment, across all times. These stronger properties allow us to achieve our theoretical guarantees, but also result in a more challenging implementation. Nevertheless, implementation is quite feasible, as demonstrated in our empirical evaluations.
>
> [1] Woodruff, David, Peilin Zhong, and Samson Zhou. "Near-Optimal k-Clustering in the Sliding Window Model." Advances in Neural Information Processing Systems 36 (2024).
>
> > The paper lacks a sufficient number of comparison algorithms, making the experimental results less convincing. Additionally, the parameter choices and values for $\alpha$ and $\beta$ are not specified, limiting the reproducibility and clarity of the experimental parts.
>
> > Q3:… The authors should add more sliding window algorithms, fair algorithms or heuristic algorithms as comparisons to make the experimental parts better.
>
> Thanks for the suggestion. We have conducted additional empirical evaluations, incorporating both larger datasets and new baselines. In particular, we evaluated both (1) an algorithm for clustering in the sliding window model that does not consider fairness (Borassi et al. 2020) and (2) an algorithm for fair clustering in the offline setting that does not consider the sliding window model (Backurs et al., 2019). Our experiments demonstrate that our algorithm performs better than (Borassi et al. 2020) in accuracy (using similar space and time), and achieves a comparable objective value while using much lower space than (Backurs et al., 2019) which needs to store the whole sliding window.
>
> > Q4: How to determine the value of parameter $k$? Does different choices of $k$ influence the experimental results of the proposed algorithms? What are the choices of the values for $\alpha$ and $\beta$.
>
> We do not pick $k$ specifically for every dataset, and we use the same $k = 10$. It is certainly better to pick $k$ according to the cluster structure of each dataset, but this is probably not necessary for our purpose, which is to evaluate the objective value/running time between baselines in a consistent way.
>
> A larger $k$ may require us to set a larger coreset size in order to achieve a similar accuracy, but we do not expect this coreset size to increase very fast with respect to $k$ (and recall that the coreset size is a small polynomial of $k$ even in the worst case). We are still looking at additional experiments for varying $k$ in the next version.
>
> In all our experiments, we use $\alpha = 0.1$ and $\beta = 0.9$. Again, we do not try to pick these specifically according to the structure of the dataset, and we simply use some typical value that is consistent between baselines.

---

> > ### Comment · Reviewer_hpoL · 2024-11-25
> > **Response to the Rebuttal**
> >
> > Thank you for the response. I would like to maintain my initial evaluation.

---

### Author Response · Authors · 2024-11-22
**Response to all reviewers**

We appreciate the reviewers for their insightful comments and constructive feedback, as well as their positive remarks, including:
- The theoretical results of the paper are solid. (Reviewer hpoL)
- This paper establishes lower bounds for the fair clustering in sliding window model. (Reviewer hpoL)
- The proposed method achieves near-optimal clustering performances with $(1+\varepsilon)$-approximation on clustering quality guarantees while the space complexity nearly matches the lower bound provided. (Reviewer hpoL)
- The proposed method uses sublinear space in sliding window model, which can be used for handling large, dynamic datasets that require efficient memory usage. (Reviewer hpoL)
- Complementing these theoretical findings, this paper presents empirical results on real-world datasets, demonstrating that their approach significantly improves clustering cost and stability over a uniform sampling baseline, supporting the algorithm’s practical effectiveness in dynamic, fairness-sensitive applications. (Reviewer cQ32)
- The paper identifies a crucial separation between fair clustering in different streaming models, offering new theoretical insights into the limitations of fair clustering in the sliding window model without any fairness violations. (Reviewer cQ32)
- The lower bound result is strong and well-supported by communication complexity arguments, which can have positive impact on the theoretical understanding of fair clustering. (Reviewer cQ32)
- The proposed approximation algorithm provides a $(1+\varepsilon)$-approximation with sublinear space, which is a valuable contribution in the context of fair clustering and sliding windows. (Reviewer cQ32)
- The algorithms, particularly the coreset construction, are explained with clarity, and the theoretical proofs, while complex, are well-structured. (Reviewer cQ32)
- The paper studies an important problem with pratical impacts. (Reviewer RZvP)
- It not only presents novel theoretical insights but also implemented the algorithm. The evaluation showcased that the proposed algorithm consistently achieves better results compared to a uniform sampling baseline. (Reviewer RZvP)
- This paper addresses an open theoretical problem - whether the proportional fairness constraint and the sliding window model are compatible in sublinear space, by providing the inapproximability result. (Reviewer 5dNf)
- The proposed algorithm is the first known one with an approximation guarantee for fair clustering in the sliding window model. (Reviewer 5dNf)
- The theoretical results are thorough and solid. (Reviewer 5dNf)
- The paper is generally well-written and organized. (Reviewer 5dNf)

We have conducted a number of additional empirical evaluations. In particular, we:

- Added a larger high dimensional dataset Census
- Added a new baseline algorithm from Borassi et al., 2020 to our existing experiments (sliding window algorithm that does not consider fairness)
- Added a new baseline algorithm from Backurs et al., 2019 to our existing experiments (fair algorithm that does not consider the sliding window model)
- Added an evaluation of the total running time

The new cost and running time evaluation results are available at https://github.com/13627836/ICLR25-submission9108-rebuttal/blob/main/cost_evaluation.png and https://github.com/13627836/ICLR25-submission9108-rebuttal/blob/main/total_running_time.png, respectively.

Below, we address each reviewer's initial comments separately. We hope our responses cover all raised points and are happy to answer any further questions during the discussion phase!

---

### Meta-Review · Area_Chair_31Lc · 2024-12-15

**Metareview:**

The paper considers a fair clustering problem in the sliding window model, and provides algorithms that nearly matches the lower bound. Given the theoretical contributions, and the importance of fair clustering, I recommend acceptance.

**Additional Comments On Reviewer Discussion:**

Rebuttal was sufficient.

---

### Decision · Program_Chairs · 2025-01-22

Accept (Spotlight)